# A network-based detection scheme for the jet stream core

Sonja Molnos[1,2], Tarek Mamdouh[3], Stefan Petri[1], Thomas Nocke[1], Tino Weinkauf[4], Dim Coumou[1,5]

[1] Potsdam Institute for Climate Impact Research, Potsdam, Germany
[2] Department of Physics, University of Potsdam, Germany
[3] Ain Shams University, Egypt
[4] KTH Royal Institute of Technology, Sweden
[5] Institute for Environmental Studies (IVM), VU University Amsterdam

*Correspondence to*: S. Molnos(molnos@pik-potsdam.de)

**Abstract.** The polar and subtropical jet streams are strong upper-level winds with a crucial influence on weather throughout the northern hemisphere mid-latitudes. In particular, the polar jet is located between cold Arctic air to the North and warmer sub-tropical air to the South. Strongly meandering states therefore often lead to extreme surface weather.

Some algorithms exist which can detect the 2D (latitude and longitude) jets' core around the hemisphere, but all of them use a minimal threshold to determine the subtropical and polar jet stream. This is particularly problematic for the polar jet stream whose wind velocities can change rapidly from very weak to very high values and vice versa.

We develop a network-based scheme using Dijkstra's shortest path algorithm to detect the polar and subtropical jet stream core. This algorithm considers not only the commonly used wind strength for core detection but also takes wind direction and climatological latitudinal position into account. Furthermore, it distinguishes between polar and subtropical jet, and between separate and merged jet states.

The parameter values of the detection scheme are optimized using simulated annealing and a skill function that accounts for the zonal-mean jet stream position (Rikus, 2015). After the successful optimization process we apply our scheme to reanalysis data covering 1979 - 2015 and calculate seasonal-mean probabilistic maps and trends in wind strength and position of jet streams.

We present longitudinally defined probability distributions of the positions for both jets for all on the Northern Hemisphere seasons. This shows that winter is characterized by two well separated jets over Europe and Asia (ca. 20°W to 140°E). In contrast, summer normally has a single merged jet over the western hemisphere but can have both merged and separated jet states in the eastern hemisphere.

With this algorithm it is possible to investigate the position of the jets' cores around the hemisphere and it is therefore well suitable to analyse jet stream patterns in observations and models, enabling more-advanced model validation.

**Keywords:**

Upper tropospheric jets, detection scheme, probability analysis of the jet stream cores

# 1 Introduction

Jet streams are upper-level fast currents of air that circulate and meander around the hemisphere and play a key role in the general circulation of the atmosphere as well as in generating weather conditions throughout the northern hemisphere mid-latitudes. In general, we distinguish between two jet stream types in the troposphere: the subtropical jet stream (STJ) and the polar front jet stream or, simply, the polar jet stream (PFJ).

The STJ is located at the upper branch of the Hadley circulation and forms due to momentum conservation, when air moves poleward, and meridional contrasts in solar heating (Woollings et al., 2010). The PFJ is situated along the polar front and is driven by baroclinic eddies that evolve due to temperature gradients along the region of the polar front (Pena-Ortiz et al., 2013) and is therefore often referred to as eddy-driven jet. Those transient eddies transport heat and vorticity and thereby accelerate the westerly winds (Woollings, 2010). The hemispheric north-south temperature gradient is strongest in winter and weakest in summer and this can explain variations in the jet stream strength and position between seasons. In summer, the winds are weaker and the jets move farther polewards, whereas in winter the winds are stronger and the jets move farther equatorwards as the cold front extends into subtropical regions (Ahrens, 2012).

Jet streams are thus sensible to changes in temperature gradient and variability and hence also to climate change (Barnes and Polvani, 2013; Grise and Polvani, 2014; Solomon and Polvani, 2016). Large-scale undulations in the jets (Rossby waves) can sometimes become quasi-stationary (i.e. stagnant) which can lead to persistent weather conditions at the surface. Persistent weather can favour some type of extreme weather events (Coumou et al., 2014; Stadtherr et al., 2016). Petoukhov et al. (2013) proposed a mechanism that could provoke such weather extremes in the northern hemisphere mid-latitudes. Quasi-stationary Rossby waves in summer are linked to persistent heat waves and severe floods (Kornhuber et al., 2016; Petoukhov et al., 2013, 2016). Likewise in winter, strongly meandering jets, driven either by anomalous tropical (Palmer, 2014; Trenberth et al., 2014) or extratropical (Peings and Magnusdottir, 2014) sea-surface temperatures or stratospheric variability (Cohen et al., 2014; Kretschmer et al., 2016), can lead to mid-latitude cold spells.

Hence, jet streams play a key role in the general circulation and for generating mid-latitude weather conditions and extremes. Several schemes have been proposed to extract the jet stream positions from wind data, each one with advantages, but also limitations.

Rikus developed a detection-method to analyse zonal mean positions of the jet streams (Rikus, 2015) using the zonally averaged zonal wind in latitude – height space to identify local maxima as cores of the jet streams. This method thus cannot analyse the development of the jet stream in the longitudinal East-West direction.

A method for calculating the jet stream core in latitude-longitude-direction was developed by Archer and Caldeira (2008). They define the jet's latitudinal position for each longitude using mass-flux weighted monthly mean wind speed between 100 and 400hPa in the northern ( 15N - 70N) and southern hemispheres (SHJ: 40S - 15S; SHP: 70S - 40S ).

Their algorithm detects only one jet position in the northern hemisphere and thus cannot distinguish between polar and subtropical jet streams. It is also not possible to capture omega-shaped jet patterns, since that method assigns only one latitude for each longitude.

Koch et al.(2006) classify so-called deep or shallow jet stream events. Their three-step algorithm first calculates the vertically averaged horizontal wind speed between two pressure levels ($p1 = 100$ hPa and $p2 = 400$ hPa) for each time instance and grid point. Next, a threshold of $30$ ms$^{-1}$ is applied to detect a so-called jet event in a grid cell. Further analysis over vertical layers classifies events into deep or shallow jet stream events but it does neither extract the actual stream core, nor does it distinguish between polar and subtropical jet stream (Koch et al., 2006).

Gallego et al. developed a scheme using a geostrophic streamline of maximum daily averaged velocity at 200 hPa to find the jet stream in the southern hemisphere. It uses wind velocities threshold of $30$ m s$^{-1}$ and distinguishes between the subtropical and polar jet stream, when the average latitudinal difference is greater than 15°. The threshold was set by manual optimization (Gallego et al., 2005). This approach might work reasonably for the southern hemisphere jets, a fixed threshold approach is particular problematic for the northern hemisphere polar jet which can change drastically in strength on weekly timescales.

The first 3D method (longitude, latitude, height) developed by Limbach at al. (2012), detects and tracks specific properties of atmospheric features as merging and splitting jet streams (via clustering of data points). Still this method cannot distinguish between subtropical and polar jet streams and also requires the use of a wind velocity threshold (Limbach et al., 2012).

Another 3D detection scheme was developed by Pena-Ortiz et al. (2013), which identifies local wind maxima in the zonal wind field by using a specified wind speed threshold. The algorithm distinguishes between the subtropical and polar jet stream via a specified threshold in latitude. A limitation of such an approach is that the values of such thresholds are not well defined. In particular the polar jet, which is our prime interest, can meander over large latitudinal ranges and experience strong variability in its strength (Pena-Ortiz et al., 2013).

To overcome these issues, we propose a new method which uses a Dijkstra's Shortest Path algorithm to find the shortest path in a network of nodes and edges with an edge cost function, defined by any combination of relevant variables. We develop a 2D detection scheme for both the PFJ and STJ core, and define our edge cost function using wind speed, wind direction, and a latitudinal guidance parameter (which is not thresholded). This way, we are able to accurately differentiate between subtropical and polar jet.

In section 2 we describe the data used in this algorithm. In section 3 we explain the details of our detection scheme, parameter optimization process and its results. Afterwards (section 4), we analyse jet stream positions since 1979 and calculate probabilistic maps for different seasons. In section 5, we calculate trends in latitudinal position and wind strength for the STJ and the PFJ. We conclude with a summary and a discussion in section 6.

## 2 Data

In this study, we used ERA-Interim data (Dee et al., 2011) from the European Centre for Medium-Range Weather Forecasts (ECMWF). The ECMWF provides meridional and zonal wind velocity components with a 0.75 latitude-longitude grid resolution. We chose 11 vertical layers of the upper troposphere stretching from 500 mb to 150 mb and for four 6-hourly timesteps per day (0:00h, 6:00h, 12:00h, 18:00h) for the years 1979–2014. From this data, we calculate 15 - day running mean and vertically averaged (mass-weighted) wind velocity, which is used for all analysis in this paper.

In the following text, a "time period" denotes a 15 - day mean centred on a given day.

## 3 Methods

Our jet stream core detection scheme is based on Dijkstra's shortest path algorithm, which is a widely used method for finding the shortest path from a source to a destination within an edge-weighted graph (Dijkstra, 1959). We assume that the jet stream core is a closed path along the hemisphere, source (most westerly point) and destination (most easterly point) are at the same location.

We use wind data on a 2-dimensional grid of the northern hemisphere where each grid point is taken as a node in a network graph. Only geographically adjacent grid points respectively nodes are connected via edges and thus no teleconnections are considered. The nodes within the most westerly column are copied after the end of the most easterly column to ensure that that the path found with Dijkstra's algorithm starts and ends at the same location. The path itself is not an injective function of longitude meaning that the path can pass multiple times the same longitudinal coordinates.

To avoid noise and reduce computational costs only those grid points where the wind velocity is greater than 10% of the maximum wind velocity for the considered time period are connected.

In order to reduce computational costs, the spatial domain is reduced to the main region of interest 0°–75°N for the subtropical jet stream on the northern hemisphere. The spatial domain for the polar jet stream is 0°N–90°N, since in some rare cases the polar jet stream could be occasionally close to the 30∘N limit.

We define an edge cost function $C_j$ based upon wind speed, wind direction and a latitudinal guidance-function using the climatological mean latitudinal position of each jet:

$$C_j = w_1 X_j + w_2 Y_j + w_3 Z_j,$$

$$w_1 + w_2 + w_3 = 1 \tag{1}$$

The variables $X_j, Y_j$ and $Z_j$, each normalized to the interval [0,1], are the three terms for computation of the cost at edge $e_j$ and $w_1, w_2$ and $w_3$ are the weights that control the contributions of the three cost terms. These weights are non-negative and

their sum is equal to one.

The three terms and their respective factors are illustrated in Fig. 1(a-b). Figure 1(a) shows all nodes and edges as well as the wind velocities of the considered node (blue arrows) in the grid. For each edge $e_j$, its cost is computed depending on the wind velocities (term $X_j$, length of blue arrows) and wind directions (term $Y_j$, angle between blue arrow and black edge) at its two nodes A and B and from its latitude (term $Z_j$, shown in Fig. 1(b)).

The first term $X_j$ captures the magnitude of the wind field at the nodes A and B. Jet streams are strong upper-level winds and hence the jet stream core should be there, where the wind strength is maximal:

$$X_j = 1 - \frac{\sqrt{u_A{}^2 + v_A{}^2} + \sqrt{u_B{}^2 + v_B{}^2}}{2 \max_{k=1}^{n}(\sqrt{u_k{}^2 + v_k{}^2})}, \tag{2}$$

whereby $u_A, u_B, v_A$ and $v_A$ are the zonal and meridional wind speeds at nodes A and B connected by edge $j$ and $\max_{k=1}^{n}(\sqrt{u_k{}^2 + v_k{}^2})$ is the maximum wind speed found at the considered time period for any node $k$ (see also Fig. 1(a)). The second term in Eq.(2) is thus always smaller than or equal to 1. We subtract this value from 1, and thus low values of $X_j$ refer to high wind speeds, because Dijkstra's algorithm will minimize the edge cost of the path (i.e. find the shortest path).

The second term $Y_j$ weights each edge $e_j$ according to the angle between the normal vector of the edge and the wind direction:

$$Y_j = \frac{1 - |V_A| \cdot |e_j|}{2} \tag{3}$$

Here $|V_A|$ is the normalized vector of the wind direction in node $A$ and $|e_j|$ is the normalized vector of the edge direction (see also Fig. 1(a)).

The third term $Z_j$ is used to differentiate between Polar and Subtropical jet streams. Basically, it favours pathways that are close to the climatological mean latitude of polar and subtropical jet but still allows free movement within a latitudinal belt of roughly $\pm 20\%$ of the climatological mean. Outside this latitudinal belt, $Z$ rapidly grows according to

$$Z_j = \frac{(\phi_j - \phi_{\text{clim}})^4}{[\max(\phi_{\text{clim}}, 90 - \phi_{\text{clim}})]^4} \tag{4}$$

Hereby, $\phi_j$ and $\phi_{\text{clim}}$ are the latitude of the edge and of the climatological mean latitude, respectively.
The reason for taking the difference between the latitudes raised to the power 4 is to give flexibility to the detected path to move almost freely in the vicinity of the desired latitude, but a strongly increasing weight farther away. This is also illustrated in Fig. 1(b), where the condition $Z_j$ for the STJ and PFJ is shown.

==Naturally, there are other== slightly different ways to define wind strength, wind direction and latitudinal dependence for the edges of the network. For example, $X_j$ and $Y_j$ could be merged to a term, which considers the wind projection along the edge unitary vector. In addition, it is possible to use a lower- or higher ordered function for equation 4, e.g. a linear function or a function with the order of 8. However, a lower order means less free movement within the latitudinal belt centered around

$\phi_{\text{clim}}$. A higher order has negligible effects since Eq. (4) within the central latitudinal belt already gives values close to zero. After calculating the edge cost for each edge according to Eq.(1), our algorithm returns from the set of all possible paths $P_i$ with total edge costs of the path $TC_i$ the path $P_{min}$ with minimal total edge cost $TC_{\text{min}}$:

$$TC_{\text{Min}} = \text{Min}(TC_i) = \text{Min}\left(\sum_{j=0}^{n} C_j,\right) \tag{5}$$

where $n$ is the number of edges in the path $P_i$.

## 3.1 Calibration of weights

The optimal weights $w_1, w_2$ and $w_3$ and the climatological latitude $\phi_{\text{clim}}$ are determined with a calibration scheme using Simulated Annealing and Rikus' algorithm.

Rikus' algorithm is a closed contour object identification scheme (Rikus, 2015). It operates on a zonal mean zonal wind and treats the two dimensional (pressure height and latitude) zonal mean U field for every time period as a single isolated image using image coordinates defined by $x$- and $y$-position.

Figure 3 shows the scheme of Rikus' algorithm. First a local maximum (minimum) filter is applied to the original zonal mean U field. The maximum (minimum) filter is defined as a 25 point maximum stencil (25 point minimum stencil) applied to the total U field. The stencil algorithm replaces the maximum (minimum) value within a box of 5 points in $x$- and $y$-direction (resulting in a total 25 grid points) to the central grid point of that box. The box with the central grid point $(x, y)$ moves over the total U field starting at the upper left corner of the zonal mean U field and ending at the at the lower right

corner.

This way the fields $U_{\text{Min}}$ and $U_{\text{Max}}$ are determined (Fig. 3 b, c).

In a second step Rikus' algorithm examines for each grid cell whether $U_{\text{Max}}(x, y) - U_{\text{Min}}(x, y) > 0.4$ and whether $U_{\text{Max}}(x, y) = U(x, y)$ (Fig. 4 d, e). Only points where both conditions are fulfilled are zonal mean jet stream cores (Fig. 3f, blue points).

We applied Rikus' algorithm to the zonal mean zonal wind field of each time period (i.e. 15 - days running mean ERA-Interim data (Dee et al., 2011)) to identify the zonal mean jet stream latitude for all levels and latitudes in the domain 150mb–430mb and 50°N–70°N (15°N–50°N) for the years 1979–2014. We selected those days, where one polar and/or one

jet stream within the above mentioned region were found. We used Rikus' algorithm in a skill function to be minimized with simulated annealing to calibrate the weights of Eq. (1).

Simulated Annealing (Kirkpatrick, 1984) is an optimization method that approximates the global minimum of a high-dimensional skill score function. We use the multi-run simulation environment SimEnv (Flechsig et al., 2013), to calibrate the weights $w_1$ and $w_3$ as well as $\phi_{\mathrm{clim}}$ of Eq. (1) and (4) for the PFJ and STJ separately. We define the skill function such that our results in the zonal-mean match those of Rikus' algorithm.

We expect the mean of all latitudinal positions calculated by our algorithm to be close to the zonal mean jet position found by Rikus' algorithm and thus define our zonal mean skill function accordingly:

$$S = \sum_{t=1}^{t_{\mathrm{end}}} \sqrt{[\phi_{\mathrm{Rikus}}(t) - \phi_{\mathrm{mean}}(t)]^2}, \qquad (6)$$

where $\phi_{\mathrm{mean}}(t)$ is the zonal mean of all latitudes found by our algorithm, $\phi_{\mathrm{Rikus}}(t)$ is the zonal mean latitude of the jet stream core determined by Rikus' algorithm. We take the sum of the differences in latitude for all time periods $t$, where Rikus'algorithm finds a jet core ($t_{\mathrm{end}}$ is the number of such time periods). The scheme is illustrated in Fig. 2.

The reason for tuning our spatially resolved tool to a zonal mean approach is that the characteristics of the jet stream like the zonal mean latitude position should be ultimately the same. The mean latitude detected by our algorithm should be very close to the maxima in zonal mean zonal wind.

We determined the wind direction weight $w_2$ manually, since it only smooths the curve locally and therefore does not affect the zonal mean position used for tuning. For the manual tuning of $w_2$, we tried different values for different time periods and found a value of 0.0015 to give the most desirable results. Since this weighting factor only affects local smoothing, its value does not affect the hemispheric path found.

As starting point for our automatic optimization scheme the parameters ($w_1, w_3$ and $\phi_{\mathrm{clim}}$) of the graph for Dijkstra's algorithm were set to manually selected values as listed in Table 1. We chose the parameters $w_1$ and $w_3$ such that both parameters have approximately the same value. For $\phi_{\mathrm{clim}}$ we chose the known climatology value for STJ and PFJ respectively (Ahrens, 2012). Since the position of the jets changes depending on season, we allow our algorithm to alter this parameter.

With the zonal mean subtropical and polar jet stream latitudes found by Rikus' algorithm we optimized the parameters $w_1, w_3$ and $\phi_{\mathrm{clim}}$ for cold (November, December, January, February, March, April) and warm months (May, June, July, August, September, October). For computational reasons, we first optimize the STJ parameters using every 14$^{\mathrm{th}}$ time period. This first step gives us proper starting conditions for the final optimization. Thus, in the final optimization we include all

time periods and used as starting point the optimized parameters found in the first step, which strongly speeds up convergence of the annealing method. For the polar jet stream, we used all jet stream cores found by Rikus' algorithm.

## 3.2 Results of the optimization process

The results of our automatic optimization scheme are listed in Table 1. The jet stream guidance parameter $w_3$ needs to have a
strong weight in order to separate the STJ and the PFJ. This large value of $w_3$ is admissible, since Eq. (4), which describes the latitudinal guidance, gives within the central latitudinal belt values close to zero. Hence the current choice still allows free movement of roughly $\pm 20\%$ of the climatological mean.

The climatological mean latitude $\phi_{\text{clim}}$ shifts poleward in the warm season for both subtropical and polar jet, reflecting the seasonal cycle.

We would like to emphasize that all terms are important even though $w_3$ has the biggest value. If we would consider only $Z_j$, and exclude all other terms, the jet stream core would be a straight line at $\phi_{\text{clim}}$, since this would be the shortest path.
The zonal-mean latitudinal difference between Dijkstra (a longitudinal resolved latitude) and Rikus (a zonal mean latitude) for the subtropical jet stream ($<2°$) is always smaller than the difference for the polar jet stream ($<5°$). This is indeed expected as the PFJ, strongly meanders (Di Capua and Coumou, 2016), whereas the STJ is strongly zonally oriented.

Improvements in the detected jet stream core positions due to the optimization process relative to the positions found by the untuned algorithm (Fig. 4, parameters are given Table 1) are illustrated in Fig. 5. Here, the left panels show the zonal mean latitude of the jet stream core calculated with Dijkstra's algorithm (light blue lines) and that computed by Rikus' algorithm (blue circles). The black solid (dashed) lines are the borders of the PFJ (STJ) core latitudinal positions as detected with Dijkstra's algorithm around the hemisphere.

After tuning, the zonal mean latitude of the polar jet stream core detected with Dijkstra's algorithm is close to the latitude computed by Rikus' algorithm (compare Fig. 5 with Fig. 4). Moreover, visual inspection of the right panel of Fig. 5 illustrates that our algorithm now correctly finds the polar jet around the hemisphere.

The mean latitude calculated with Dijkstra's algorithm does not always match perfectly with the mean latitude computed by Rikus' algorithm, because the first is a 2D−algorithm in longitude and latitude and the latter is a 2D–algorithm in latitude
and height. Rikus' algorithm therefore does not capture the undulations of the jet stream.

Often any such differences are related to the existence of not one but two zonal-mean PFJ maxima. For example in Fig. 6 there exists a zonal mean maximum at latitude ~55°N and another maximum at ~73°N (left panel) but this is due to the undulation features of the jet stream pattern (right panel). Our algorithm resolves that undulation pattern, whereas Rikus' only detects the stronger southerly maxima, since it searches in the range between 50°N and 70°N for the polar jet stream.
For that reason its mean latitude is in-between the two maxima. Moreover our approach is able to detect a high-over-low-

blocking situation for the PFJ, in contrast to e.g. Archer and Caldeira (2008) (see Introduction).

In other cases a zonal-mean maximum found by Rikus' algorithm exists only in one longitudinal range. For example, in Fig. 7 the maximum of the pressure-height- latitude plot exists mainly because of the region between $0°E - 100°E$ and around $70°N$ latitude. Since in other parts a different path represents the polar jet stream, the mean jet stream cores are not the same.

Figure 7 shows a situation where also other paths for the STJ and the PFJ could be considered, the jets split into two jet stream cores.

In Fig. 8 the differences between the zonal mean polar jet stream cores calculated by Rikus' algorithm and with Dijkstra's algorithm are shown in two different subplots. Panel (a) shows a day-year plot depicting in blue days for which Rikus' algorithm finds a polar jet stream in agreement with the range of jet stream core latitudes detected with Dijkstra's algorithm.

In yellow are those days, where Rikus' polar jet stream core position is not between the minimum and maximum latitude of the polar jet stream path detected with Dijkstra's algorithm. These are 199 of 3122 data points which is equivalent to 6.4%. Figure 8 (b) shows the difference between the mean latitude calculated by Rikus' and the mean latitude calculated with Dijkstra's algorithm. The mean of the difference is 5°, but there are also some cases, where the difference is much higher, up to 20°. These differences are due to the undulations explained above.

The day–year plot of the subtropical jet stream in Fig. 9 shows that for every single time period Rikus' latitude position is within the range of latitudes found with Dijkstra's algorithm. Figure 9(b) indicates the difference between the mean latitude calculated by Rikus' and the mean latitude calculated with Dijkstra's algorithm, which is very small. The mean is 2° and the highest values are 6°.

## 4 Jet stream probability analysis

In this section we present some results of the analysis of the jet stream paths that were detected by our algorithm.

Figures 10, 11, 12 and 13 show probabilistic jet stream positions for different seasons with brown dashed contour lines representing the subtropical jet and black solid contour lines representing the polar jet.

The seasonal cycle of the STJ is clearly seen with winter latitudes between 20° and 40° latitudes and summer latitudes further north. Moreover, in summer the probability that the jets merge in the western hemisphere is higher, whereas in winter

the probability that they are clearly separated over almost all longitudes is higher.

In addition, the probability frequency of the PFJ is much broader than the probability of the STJ and no clear latitudinal shift between seasons is observed. In particular in summer the PFJ distribution is smeared out (indicating large fluctuations in its position) whereas in winter it is more confined.

This strong meandering of the eddy-driven PFJ is explainable due to the nature of wave-mean flow-feedbacks (Harnik et al.,

2014). The PFJ cores lie always between 40°N-80°N, only in longitudinal direction there is a seasonal dependence. Over Asia the probability of a high-latitude PFJ is larger in summer than in winter. Over Europe the probability of a low latitude

PFJ is higher in summer. This is also observable for East Pacific and America, but less pronounced, instead there seem to be in spring and summer two preferable states: merged jet state with a jet at ca. 50°N and a second state with two jets at respectively ca. 50°N and ca 70°N.

In general the probability of PFJ at low latitude is small over the European sector compared to other regions and therefore double jet states occur in every season here. In North America such a clearly separated STJ and PFJ is only observed in winter.

This coexistence of the STJ and PFJ in the eastern hemisphere, compared to more frequent merged jet states in the western hemisphere, is well documented in the literature, but was never shown in probabilistic plots as presented here (Eichelberger and Hartmann, 2007; Li and Wettstein, 2012; Son and Lee, 2005; Woollings, 2010a). Those different jet stream states occur since the processes which lead to their existence operate and interact in non-linear ways (Harnik et al., 2016; Lee and Kim, 2003). In the North Atlantic, STJ and PFJ are separated because the region of strongest baroclinicity is located relatively far poleward. In contrast, the region of strongest baroclinicity in the North Pacific is located near the latitude of maximum zonal wind, favouring a merged jet (Lee and Kim, 2003; Li and Wettstein, 2012). Such a merged jet stream is also called the eddy-thermally driven jet because of the two different genesis mechanisms. In special cases, there is the possibility that this eddy-thermally driven jet stream also appears over the North Atlantic (Harnik et al., 2014). This happens if the tropical forcing strengthens or the mid-latitude baroclinicity weakens.

In addition, the panels (b) give probabilities of the zonal-mean latitude of both jets, showing enhanced variability of the PFJ compared to the STJ. The range of overlapping latitudes between STJ and PFJ is larger in summer than in winter because of the poleward shift of the STJ. The latitudinal variability in STJ is lower in summer and winter than in spring and autumn, whereas the variability of the PFJ is similar between seasons. However, the location of the maximum in the PFJ histogram changes per season: in winter, the maximum is at ca 55°N, whereas in summer there are two maxima at 50°N and at 70°N. These two maxima probably reflect the different behaviour in western and eastern hemisphere in the PFJ. In spring, there is no clear maximum visible (between 40°N-60°N), and in autumn it is again close to 55°N.

To quantify those merged and separated states further, one could use the latitudinal difference between STJ and PFJ, for all longitudes, and this way create the probability density distributions of merged and separated jets. The presented results (Fig. 10 - 13) might in principle also be the result of clearly separated jets which displace latitudinally over time to create the overlapping probability density.

For verification, we compare the probabilistic jet fields with seasonal climatological wind fields (Panels (c)). In general, all probability density functions (PDFs) of the jet stream cores in their respective seasons coincide well with the wind fields. In summer the wind field magnitude is very low and more homogeneously spread other the hemisphere. In summer the jet stream cores are farther North than in winter due to the weaker temperature gradient in summer. In general the gradient of the wind velocities as well as the strength of the velocities in summer are weaker than in winter.

## 5 Global trends

Fig 14 shows trends in the latitudinal position and wind velocity for summer and winter as well as annual data derived from our Dijkstra –jet-detection scheme. Table 2 summarizes the results giving linear trends in mean jet stream latitude and mean wind velocity with bold values indicating statistical significance (p<0.05).

In order to compare our results with literature results, we calculated mean jet stream latitude and mean wind velocity trends, which are shown in Table 2. Bold values indicate statistical significance (p<0.05). We used Monte Carlo analysis with 10000 surrogate time series of shuffled data to determine significance (Di Capua and Coumou, 2016; Pollard and Lakhani, 1987; Schreiber and Schmitz, 2000). To account for the fact that running means present not truly independent data, we shuffle blocks of 15 days in this method.

In general, we observe a northward trend for the STJ (except for SON) which is significant for winter and annual time series. T latitudinal position of the he PFJ shows more mixed behavior with different signs for different seasons. A pronounced and significant equatorward trend is detected for the PFJ in winter. Wind velocities have generally weakened for both STJ and PFJ, something which is significant for summer, in agreement with Coumou et al. (2015) and Lehmann and Coumou (2015).

Overall these reported trends are in good agreement with previous studies though it is somewhat difficult to make direct comparisons as different studies analyzed different aspects of the flow field. For example, Pena-Ortiz et al. (2013) did not calculate separate trends for the STJ and PFJ, but instead for different range of latitudes: for winter 15°-40°, for spring and autumn 10°-70° as well as for summer 30°-60°. Since STJ winds are in general stronger, we assume that at least for spring, summer and autumn their reported trends reflect trends of the STJ. Similarly, Archer & Caldeira (2008) considered only trends in NH jet stream between 15°N-70°N, where we again expect that this mostly reflects the behavior of the STJ. Rikus (2015) calculated trends for one northern jet stream core within 20°N-54°N, so we can assume that the trend most probably describes the trend of the STJ. The findings of those studies can thus be best compared to our STJ findings. The annual poleward trend in latitudinal position of the STJ, detected with our method, is consistent with the results of Rikus and Archer & Caldeira. Also the latitudinal trend in summer calculated by our method has the same sign and order of magnitude as in Rikus and Pena-Ortiz et al, but the trend in winter is greater in our and Rikus' method compared to that from Pena-Ortiz. The trends for spring and autumn agree in sign with the analysis of Pena-Ortiz using 20[th] century data, but are weaker and even change sign for the NCEP/NCAR data set in autumn.

The wind velocity trends are positive in the publication of Pena-Ortiz, whereas we observed a negative trend as Rikus (except summer) and Archer & Caldeira. With our more-advanced approach which is able to differentiate between subtropical and polar jet, we detect stronger (and mostly significant) weakening compared to the other studies.

# 6 Summary and Discussion

We have proposed a novel and objective method to detect the subtropical and polar jet stream cores which overcomes some limitations of previous studies. Our method uses a graph approach employing Dijkstra's shortest path algorithm. With this method we are able to describe both spatially separated as well as merged jet stream cores. If the subtropical and polar jets merge, the two detected jet stream core positions become very close to each other.

We used three terms to define the edge costs: wind magnitude, wind direction and a jet stream latitudinal guidance term.

Based on those three terms, the algorithm finds the jet stream core as a closed path. Parameters entering this detection scheme were optimized using simulated annealing and comparing our spatially resolved scheme with a zonal-mean detection scheme to avoid unrealistic results. Here we discuss some possible improvements to our scheme.

Instead of using the wind direction and wind strength, it is also possible to merge both condition and consider only the wind projection along the edge unitary vector. However, with two terms we have more flexibility regarding the weights of the terms.

In addition, the jet stream latitudinal guidance term, which is in our case a fourth order function of latitude, could be a lower- or higher ordered function like a linear function or a function with the order of 8. A lower order means less freedom for the path to move away from the climatological latitude, whereas a higher order has only little effect, since the cost of a fourth order function are already small in the latitudinal belt.

As a result the latitudinal guidance term seemed the most important factor. This large value of $w_3$ is admissible, since Eq. (4), which describes the latitudinal guidance, gives within the central latitudinal belt values close to zero. Hence the current choice still allows free movement of roughly $\pm 20\%$ of the climatological mean.

We calculate the probabilities of the northern STJ and PFJ core and show that the probability of two clearly separated jet streams is very high over the East Atlantic and Eurasia and very low over the Pacific and America. This is consistent with previous studies (Li and Wettstein, 2012; Son and Lee, 2005). The underlying reason is the different location of strongest baroclinicity between the North Pacific and the North Atlantic. In the former, the strongest baroclinicity is located near the latitude of the maximum zonal wind and in the latter it is located relatively far poleward. The histograms of STJ and PFJ density for different seasons and for the annual mean show that the latitudinal variability of the PFJ is much larger than the variability of the STJ. This much larger variability is due to the nature of wave-mean flow-feedbacks (Harnik et al., 2014).

We reported the zonal-mean jet stream properties and trends of the mean latitude and wind velocity and show them to be in good agreement with other studies. Differences between studies can largely be explained by different data sets, time periods, pressure level and/or methodology (Pena-Ortiz et al., 2013; Rikus, 2015).

For future work we plan to extend the algorithm in three-dimension and apply it to the southern hemisphere. Parameters for the third dimension could be optimized in a similar way as done for latitude, but using pressure heights.

In addition, to account for splitting of the STJ and PFJ, we plan to calculate not two, but four (or even more) jet stream cores with different climatological mean latitude $\phi_{clim}$. In cases, where only one path exists, the found jet stream cores would be combined to one path (based on their similarities to each other) and in other cases, where two paths exist, they would split.

Furthermore, we intend to analyze the influence and impacts of the jet stream to extreme events using cluster analysis. This way, we can examine the link of particular cluster patterns to extreme weather events and determine which jet stream patterns have a higher probability for extremes. In addition we plan to find possible drivers which lead to those jet stream patterns, using causal effect networks (Kretschmer et al., 2016).

Another possibility is to apply our method to model data such as CMIP5 in order to analyze, whether models can reproduce the jet accurately.

**Code and data availability**

All input data was downloaded from public archives. Code and data are stored in PIK's long term archive, and are made available to interested parties on request.

**Team list**

S. Molnos, T. Mamdouh, S. Petri, T. Nocke, T. Weinkauf, D. Coumou

**Author contribution**

S. Molnos, T. Mamdouh, T. Weinkauf and D. Coumou developed the study Conception. T.Weinkauf, T. Mamdouh and T.Nocke developed the analysis method. S.Molnos, T. Mamdouh and S. Petri developed the model code and performed the simulations. S. Molnos and D. Coumou analysed and interpreted the data. S. Molnos prepared the manuscript with contributions from all co-authors.

**Competing interests**

The authors declare that they have no conflict of interest.

**Acknowledgements**

We thank ECMWF for making the ERA-Interim available. The work was supported by the German Federal Ministry of Education and Research, grant no. 01LN1304A, (S.M., D.C.). The authors gratefully acknowledge the European Regional

Development Fund (ERDF), the German Federal Ministry of Education and Research and the Land Brandenburg for supporting this project by providing resources on the high performance computer system at the Potsdam Institute for Climate Impact Research.

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

**Figures**

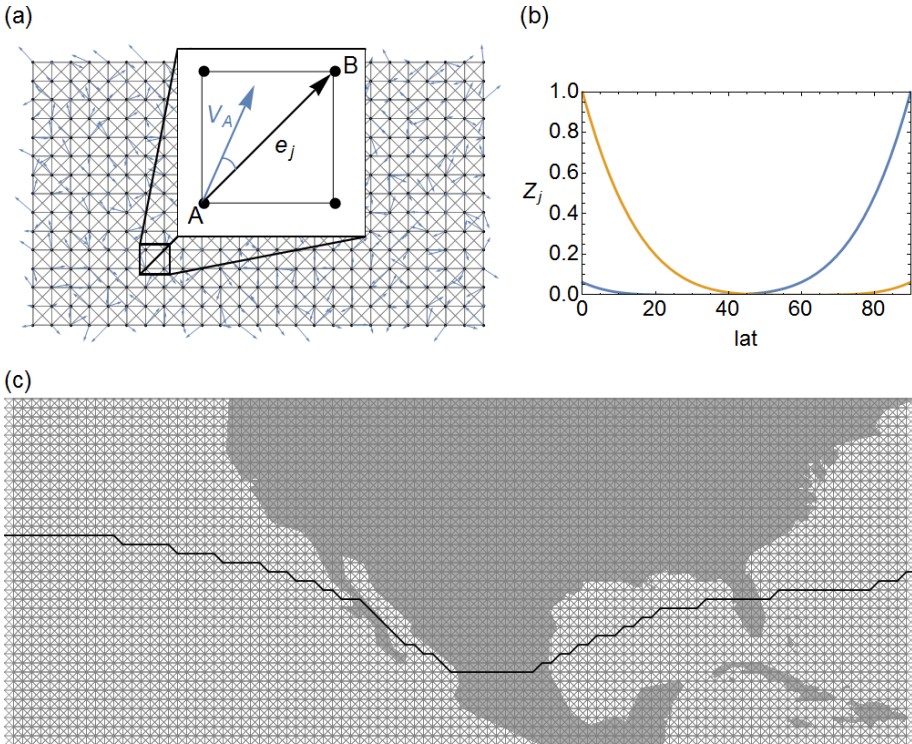

**Figure 1**. Definition of edge costs: (a) shows all nodes and edges as well as the wind velocities of the considered node (blue arrows) in the grid. The edge costs are computed from wind velocities (length of blue arrows, $X_j$), wind direction (angle between blue arrow and black edge, $Y_j$) as well as the latitudinal position $Z_j$. (b) indicates the third cost term $Z_j$ of the STJ (blue) and PFJ (orange). The edge cost is very low in the vicinity of $\phi_{clim} = 30°N$ for the STJ and $\phi_{clim} = 60°N$ for the PFJ and very high far away of $\phi_{clim}$. (c) shows the STJ (black line) in the network graph over North- and Central America for a certain time period.

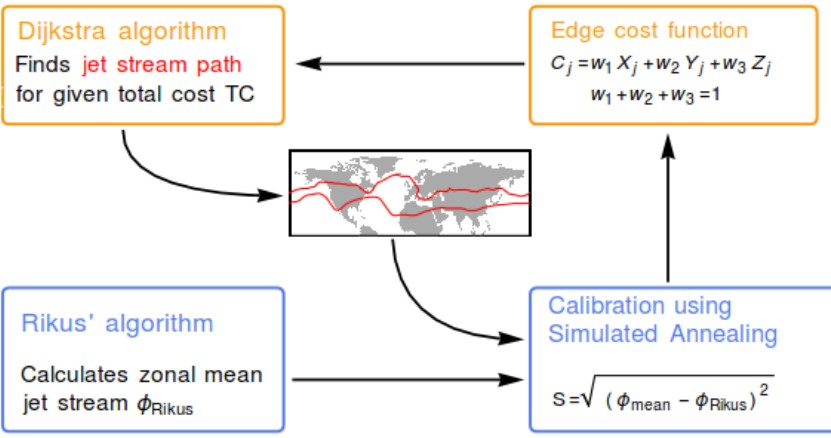

**Figure 2**. Calibration Scheme. Before calculating the shortest path with Dijkstra's algorithm, the cost of each edge has to be calculated according to the three terms $X_j, Y_j$ and $Z_j$. In order to find the correct weights of the terms, we calibrate them with Simulated Annealing and using Rikus' algorithm to construct the skill function.

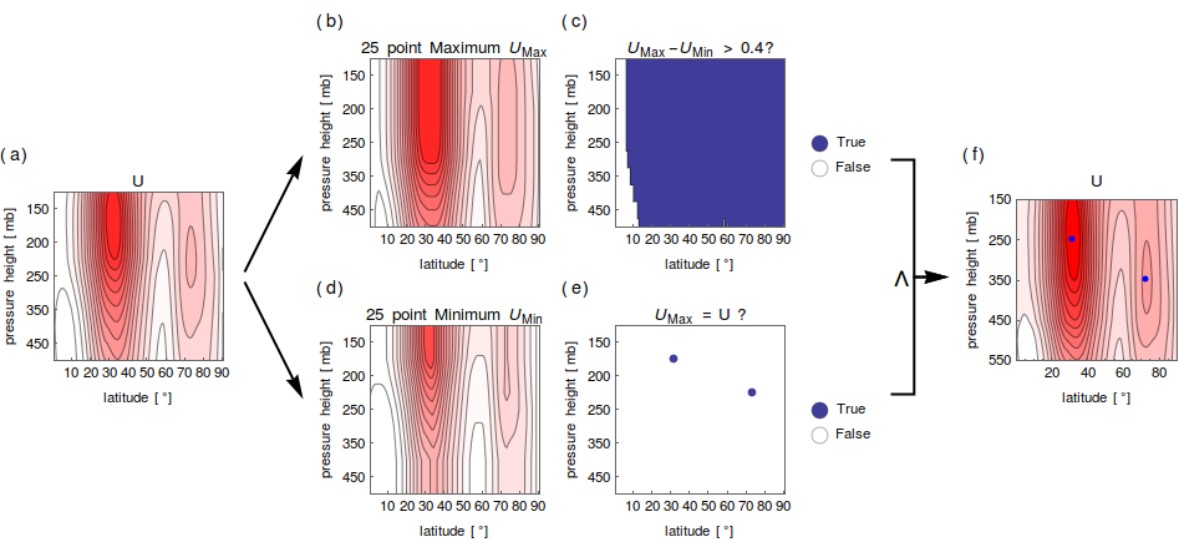

**Figure 3**. Rikus' Scheme. In (b) the 25 point maximum stencil ($U_{Max}$) and in (d) the 25 point minimum ($U_{Min}$) stencil from (a) is calculated. In (d) the condition $U_{Max}(x,y) - U_{Min}(x,y) > 0.4$ and in (e) the condition $U_{Max}(x,y) = U(x,y)$ is examined. Only those points, where both conditions are fulfilled, are zonal mean jet stream cores (f, blue points).

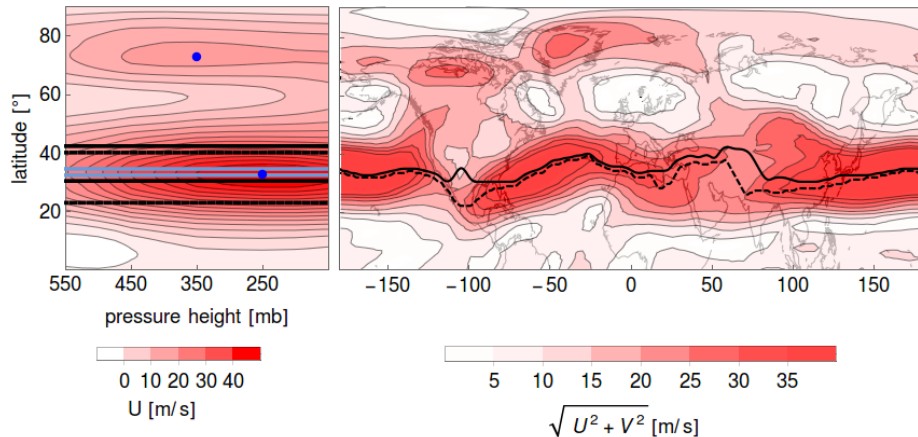

**Figure 4**. Left panel: Zonal mean latitude of the jet stream core calculated with Dijkstra's algorithm using unoptimised weights (light blue lines) and that computed with Rikus' algorithm (blue circles). The black solid (dashed) lines are the borders of the PFJ (STJ) core latitude positions as calculated with Dijkstra's algorithm. Right panel: polar (black) and subtropical (black dashed) jet stream cores are shown. (15 days running mean around 13.01.2010).

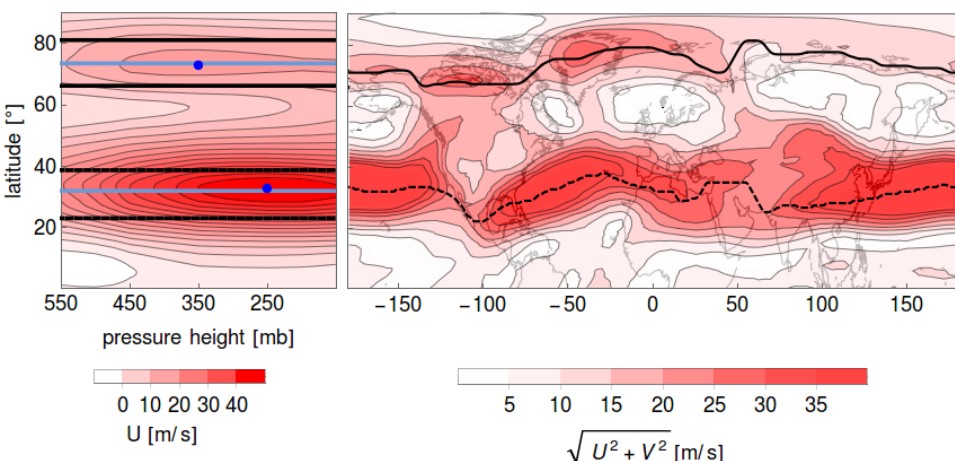

**Figure 5. 15 days running mean around 13.01.2010, jet stream cores calculated with Dijkstra's algorithm using optimized weights (compare with Fig. 2).**

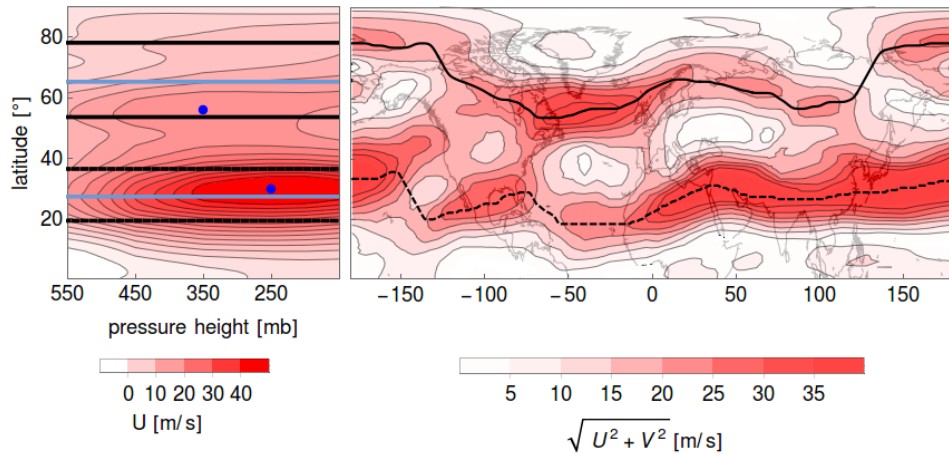

**Figure 6**. 15 days running mean around 02.03.1979. The right panel shows three maxima (30°N, 50°N and 75°N), because of that the mean jet stream core found with Dijkstra's algorithm (light blue line) does not match with the jet stream core found by Rikus' algorithm (blue circle).

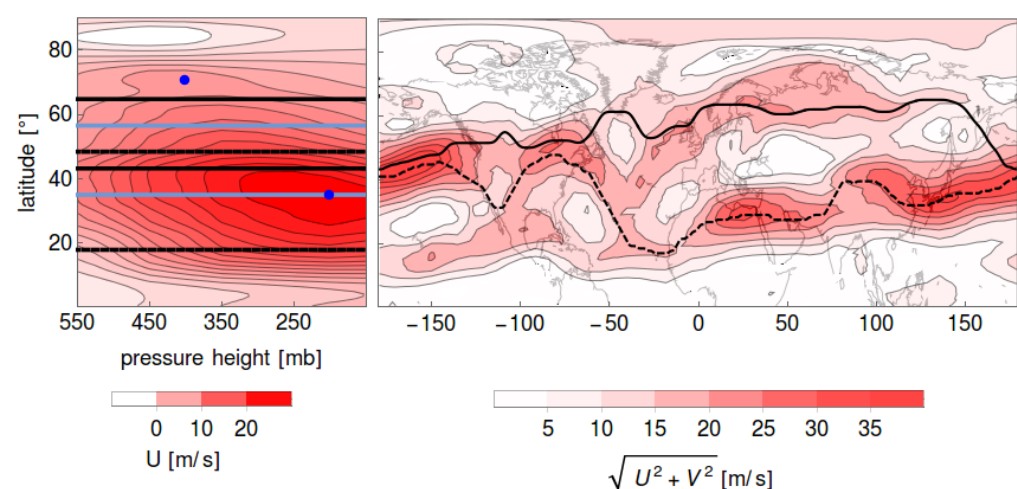

**Figure 7**. There is only a maximum in the wind field in the region between $0°E − 100°E$ and around $70°N$ latitude (15 days running mean around 12.05.1979).

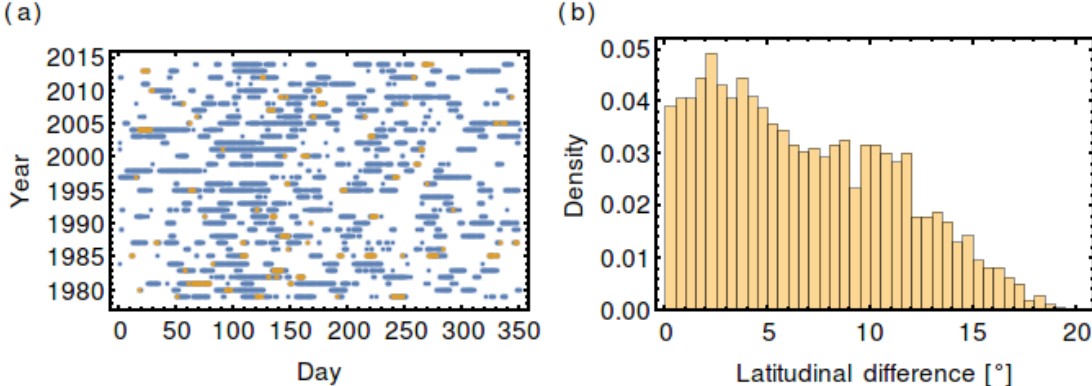

**Figure 8. (a) Day-year plot showing days used for tuning (blue) and those days, where Rikus' latitude position is not within the range of latitudes found with Dijkstra's algorithm (199 of 3122 datapoints, 6.4%), (b) Histogram of minimum latitudinal difference between the jet stream core found with Dijkstra's algorithm and the mean latitude from Rikus' algorithm, in degrees, for the polar jet stream.**

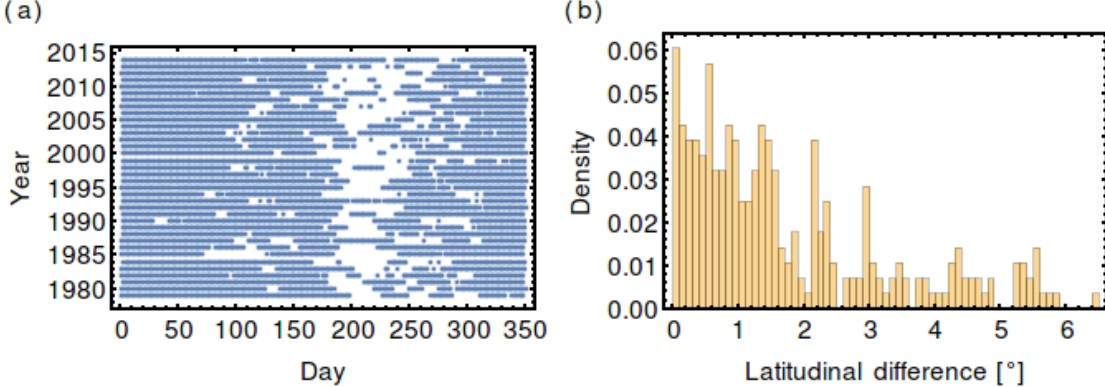

**Figure 9. (a) Day-year plot for the subtropical jet stream detection scheme (compare Fig. 8) , (b) Histogram of minimum latitudinal difference between the jet stream core found with Dijkstra's algorithm and the mean latitude from Rikus' algorithm, in degrees, for the subtropical jet stream**

.

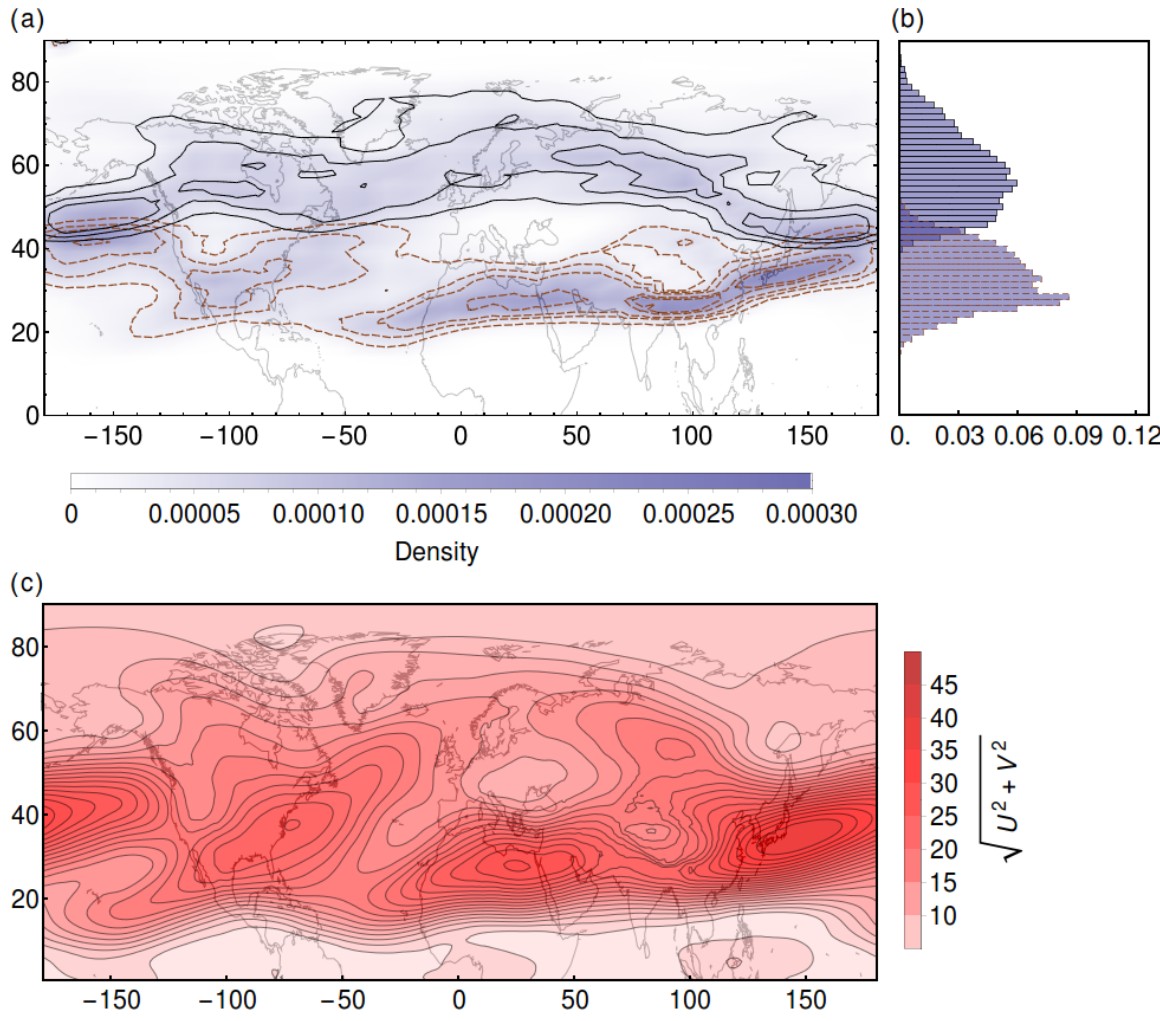

**Figure 10**. Probability analysis for spring months (MAM): (a) and (b) show the spring probability density plot and a histogram of the jet stream occurrences (1979-2014). The brown dashed contour lines represent the subtropical jet stream, whereas the black solid contour lines represent the polar jet stream. (c) depicts the climatological annual wind field (averaged over 1979-2014).

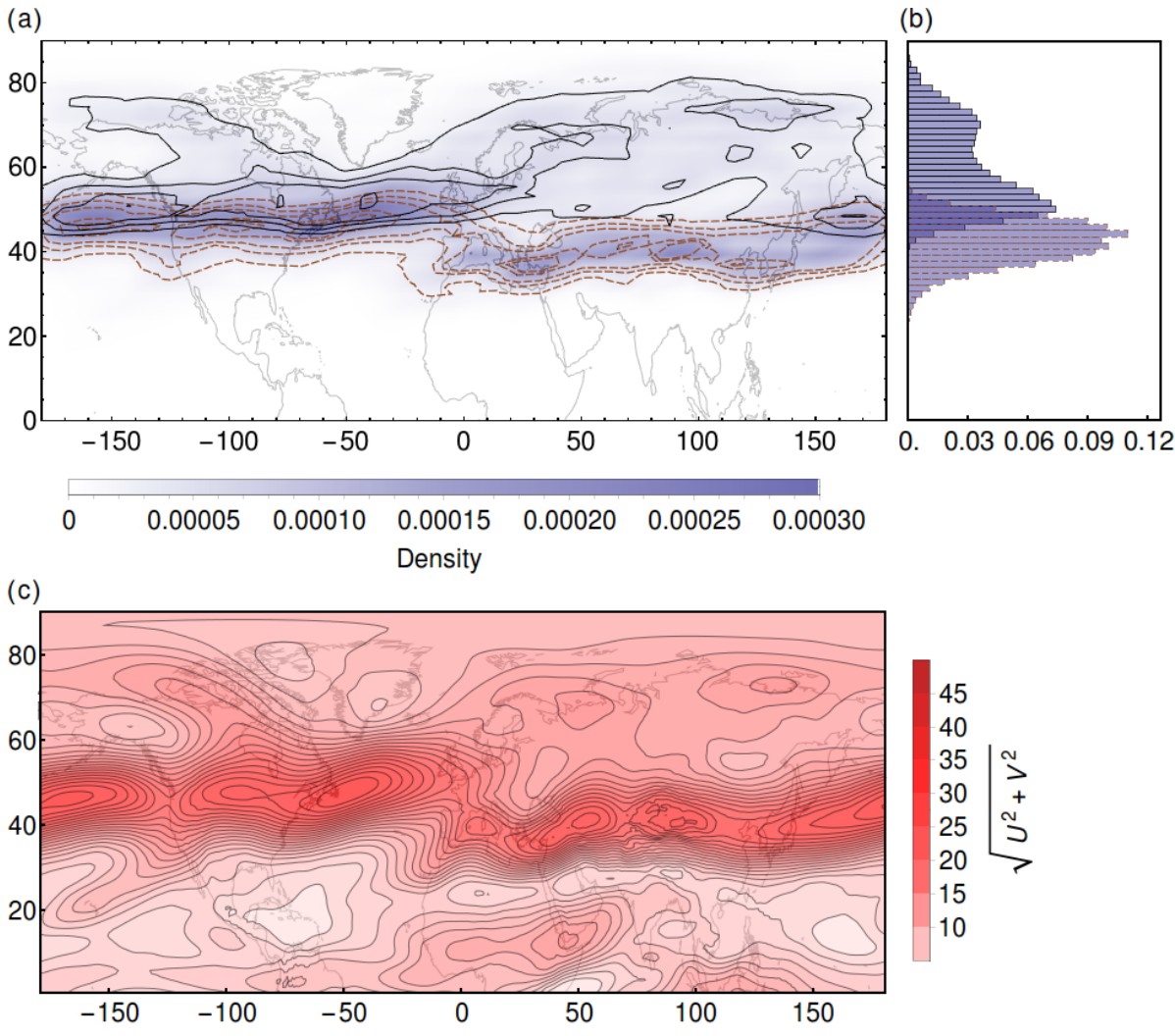

**Figure 11**. Probability analysis for summer months (JJA, compare Fig 10).

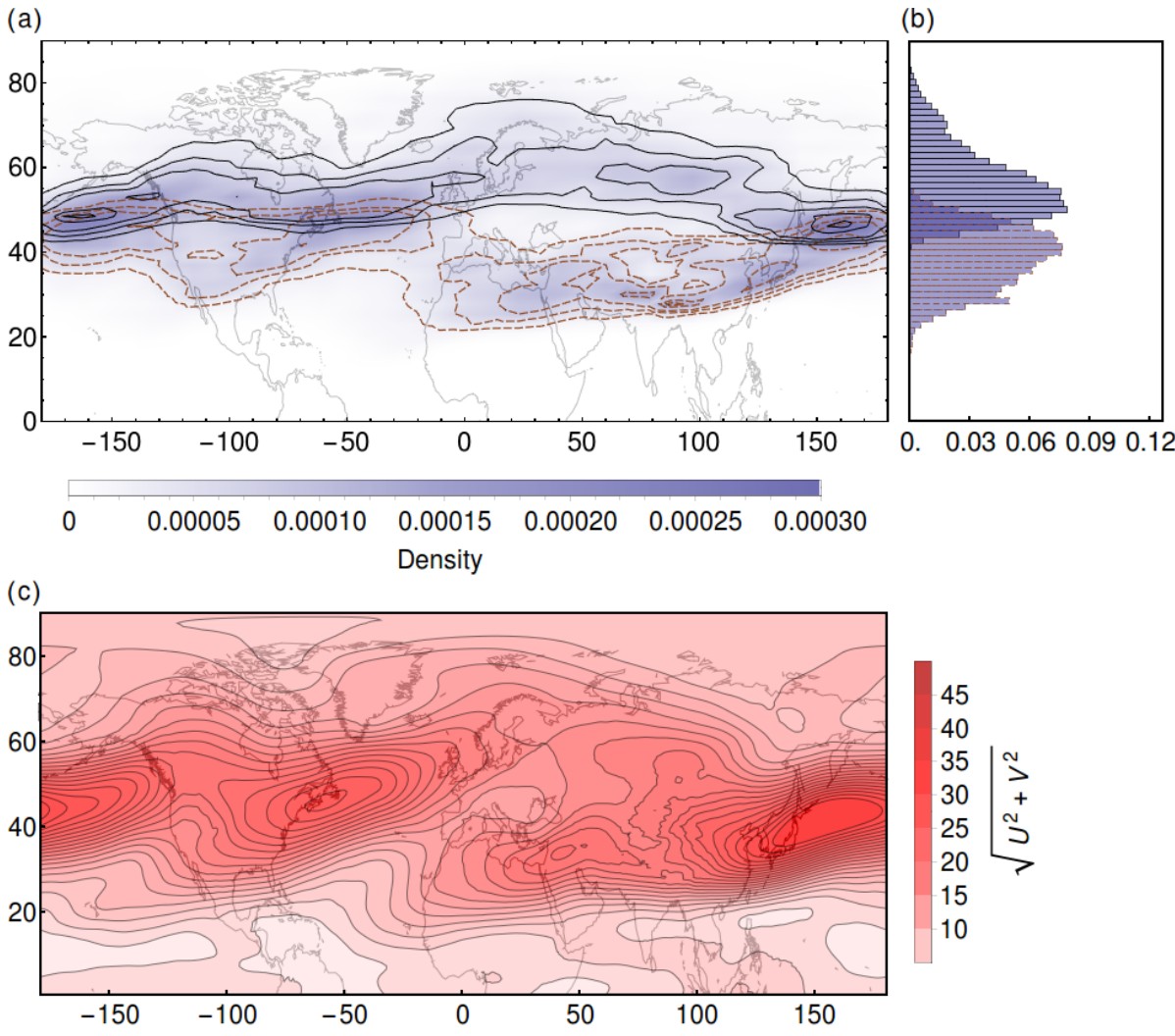

**Figure 12**. Probability analysis for autumn months (SON, compare Fig 10).

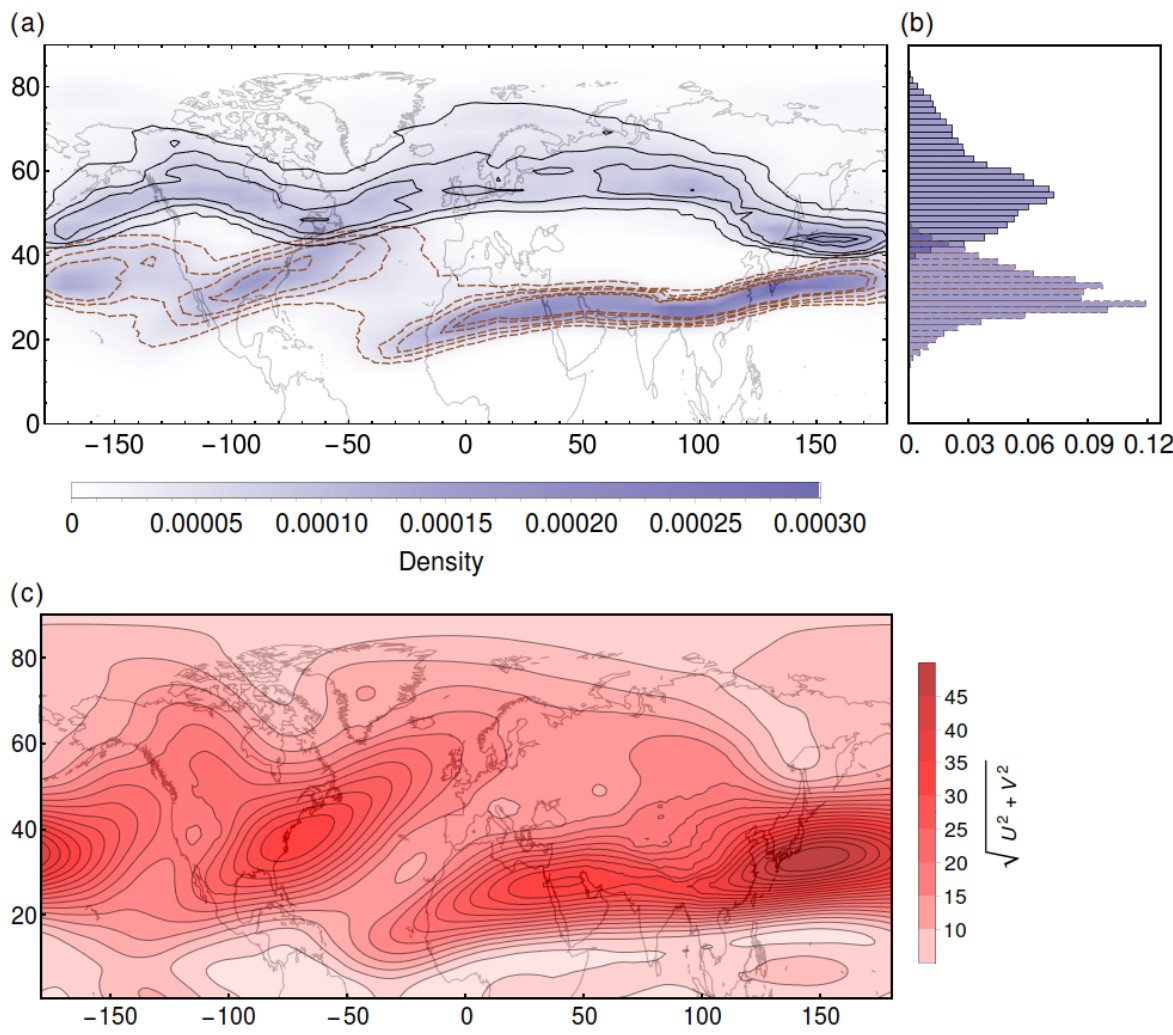

**Figure 13**. **Probability Analysis for winter months (DJF, compare Fig 10).**

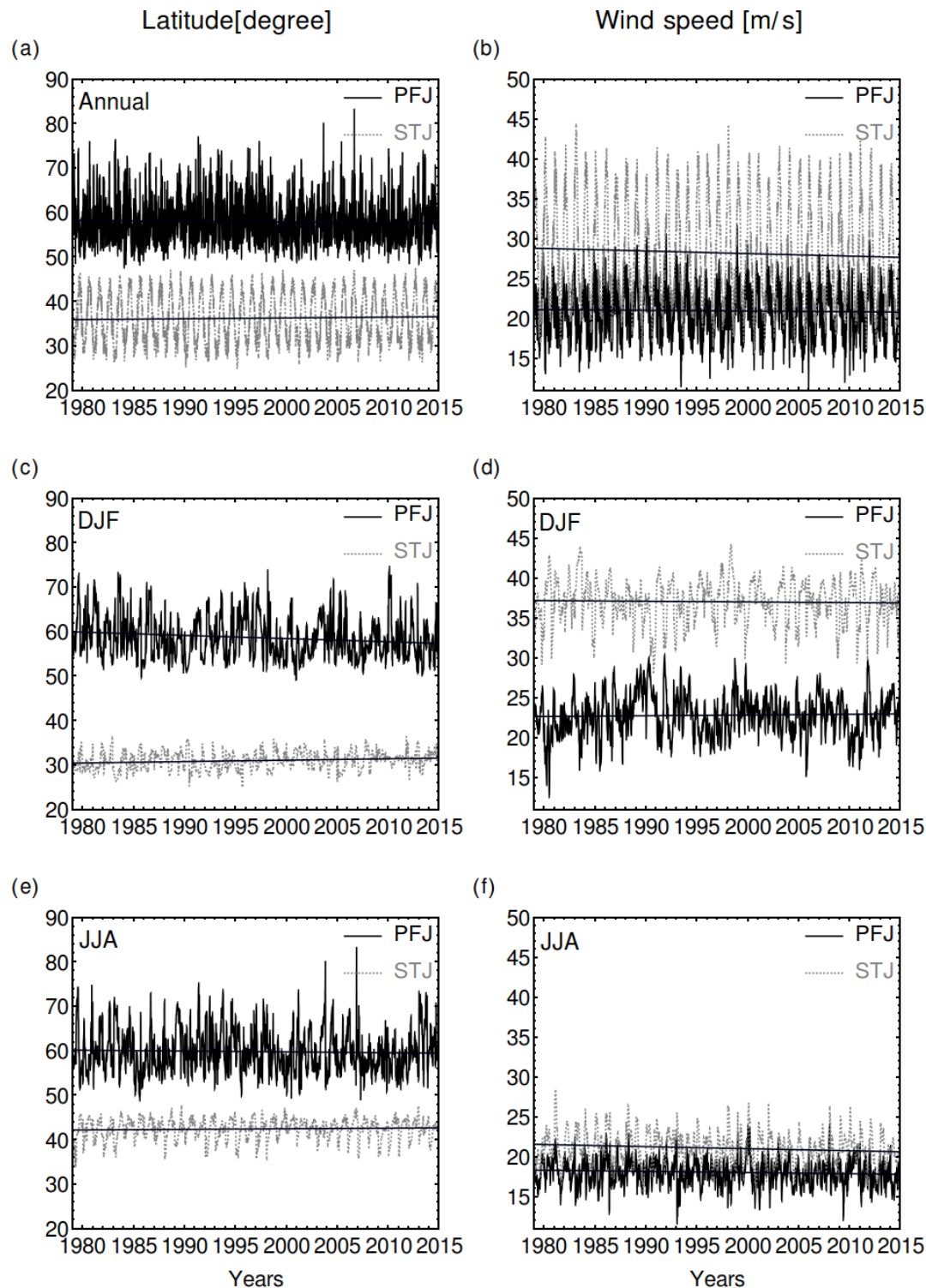

**Figure 14. Annual, DJF, and JJA: Mean latitudinal trends and mean wind velocity trends of the STJ and PFJ cores.**

**Tables**

**Table 1 Start and optimized jet stream parameters used for the edge cost function.**

| Season | Parameters | Subtropical jet stream | | Polar jet stream | |
|--------|-----------|------|-----------|------|-----------|
| | | Start | Optimized | Start | Optimized |
| cold | $w_1$ | 0.49 | 0.044 | 0.49 | 0.044 |
| | $w_2$ | 0.0015 | - | 0.0015 | - |
| | $w_3$ | 0.5 | 0.95 | 0.5 | 0.95 |
| | $\phi_{clim}$ | 30°N | 25.1°N | 60°N | 67.5°N |
| warm | $w_1$ | 0.49 | 0.072 | 0.49 | 0.043 |
| | $w_2$ | 0.0015 | - | 0.0015 | - |
| | $w_3$ | 0.5 | 0.92 | 0.5 | 0.95 |
| | $\phi_{clim}$ | 30°N | 29.8°N | 60°N | 69.1°N |

**Table 2 Slope Parameter for the latitude and velocity trends of the jet stream cores. Bold values indicate statistical significance (p<0.05) using Monte Carlo analysis with 10000 surrogate time series of shuffled data.**

| Season | Subtropical jet stream | | Polar jet stream | |
|--------|---------------------|------------------------|----------------|------------------|
| | Latitude [degree/dec] | Velocity [m/s/decade] | Latitude [deg/dec] | Velocity [m/s/dec] |
| DJF | **0.282** | -0.021 | **-0.670** | 0.061 |
| MAM | 0.244 | **-0.454** | 0.004 | -0.143 |
| JJA | 0.139 | **-0.259** | -0.189 | **-0.147** |
| SON | -0.183 | **-0.263** | 0.049 | -0.157 |
| Annual | **0.178** | **-0.321** | -0.198 | -0.085 |