# Peer review of "A network-based detection scheme for the jet stream core"

_Earth System Dynamics, 2016_

## Referee Comment (RC1) · L. Rikus (Referee) · 28 Sep 2016

General Comments:

This paper addresses scientific questions relevant to ESD because it introduces a new methodology for identification of jet streams and then applies it reanalysis data to provide a useful probability analysis of PFJ/STJ core positions. Although numerous methods to characterise the properties of jet streams exist this represents a useful contribution to the literature by introducing a novel method. One aspect of the jet literature is that although general definitions do exist there are a number of different operational definitions essentially determined by the methods used to characterise the jets, in this case local maximum wind speeds in a mass weighted layer. I thought the context of the method was a little underdone (particularly in the abstract) and the paper would

benefit from a little more discussion on other similar work even if only to highlight the advantages of this approach. Overall the method is well developed and reasonably well described but there are a few areas where the language is confusing (see comments and corrections below). I think a minimalist change to the title would probably be "A network-based detection scheme for the jet stream core". While I would suggest a couple of changes (see below) the abstract reflects the content of the paper. The overall structure of the paper is quite good although there are a couple of minor problems, e.g. Equation 6 is discussed on page 6 before it is actually given on page 7, which could be addressed. I would recommend publication with some changes.

Specific Comments:

1. I'm not so sure that algorithms to detect jet cores are lacking (as stated in the abstract). There are actually a relatively large number of previously published papers which are based on either a single level/layer or zonal/sectorial mean latitude-pressure fields. The current application of the network-based method is yet another variation of this and so needs to be put in context with other (similar) methods which use single level or mean-layer wind fields (e.g. Koch et al, 2006, Archer & Caldiera 2008, Pena-Ortiz et al 2013). Hence at the risk of expanding the paper too much I think it would be useful to acknowledge some more of the previous work and to compare with the results obtained here (even though they are based on a 15 day mean as opposed to monthly means) and discuss why this method has advantages over the previous studies.

2. I think 'time step' is a confusing choice of phrase to describe the 15 day means – maybe use 'time period' ?

3. The simulated annealing actually uses the Rikus algorithm so it is being used as more than just a comparison. (page 1, line 18) and the abstract description should reflect that.

4. Were the original runs (Fig. 2) done with with the un-optimised weights from table 1? If not what was used?

5. The supplementary plot (S1) is only mentioned in a single sentence without sufficient context to make it worth while. Either add more discussion or remove it?

Technical Corrections:

Abstract Line 21: 'mean longitudes of 20S and 140N' ?????? I don't know what this is supposed to mean!

Page 2, line 27: No year given for Gallego et al (and in reference list). Try 2005.

Page 2, line 32: there is no reference for Limbach et al (2012).

Page 3, line 2: "zonally" should be "zonal"

Page 3, line 4: "such approach" should be "such an approach"

Page 3, line 13: "all different" should be "different"

Page 3, line 18: "for 4" should be "four"

Page 4, lines1,2: I'm not sure what this line actually means!

Page 4, lines 3,4: "To avoid noise and reduce computational costs only those grid points where the wind velocity is greater than 10% of the maximum wind velocity for the considered time step are connected."

Page 4, lines 12-13: The description of the weights is inconsistent – is their sum 1 or less than 1?

Page 5, equation 4: there appear to be some brackets missing in the denominator.

Page 5, line 19: "near of 65" should be "near to 65"

Page 6, line 15: "First a maximum (minimum) filter" should be "First a local maximum (minimum) filter"

Page 7, line 5: "With the found zonal mean subtropical and polar jet stream latitudes by Rikus" should be "With the zonal mean subtropical and polar jet stream latitudes found

by Rikus' algorithm"

Page 7,lines 7-8: This sentence needs to be clarified.

Page 7, line21: "it is more undulated" should be "it undulates more"

Page 7, lines 22-23: Try - "Improvements in the jet stream core positions due to the optimization process relative to the positions found by the untuned ..."

Page 8, line 7: "polar Jet" should be "polar jet"

Page 8, line 16: "not between minimum and maximum latitude" should be "not between the minimum and maximum latitude"

Page 8, line 17: "equivalent of 6.4" should be "equivalent to 6.4"

Page 8, line 20: Change to "These differences are due to the undulations explained above." ?

Page 9, lines 16-34: The language needs to be cleaned up – the section does not scan well with a number of missing 'the' and 'a's.

Fig. 4 caption (page 15, line 14): the points in (f) are blue not white.

Fig. 5 caption (page 16, line 4): should be "compare with Fig. 2"

Fig. 6 caption (page 16, line 7): Remove ", compare Fig. 2." - it is not necessary.

References:

Gallego D, Ribera P, Garcia-Herrera R, et al. (2005) A new look for the Southern Hemisphere jet stream. Clim Dyn 24:607–621. doi: 10.1007/s00382-005-0006-7

Koch P, Wernli H, Davies HC (2006) An event-based jet-stream climatology and typology. Int J Climatol 26:283–301. doi: 10.1002/joc.1255

Limbach S, Schömer E, Wernli H (2012) Detection, tracking and event localization of jet stream features in 4-D atmospheric data. Geosci Model Dev 5:457–470. doi:

10.5194/gmd-5-457-2012

Pena-Ortiz C, Gallego D, Ribera P, et al. (2013) Observed trends in the global jet stream characteristics during the second half of the 20th century. J Geophys Res Atmos 118:2702–2713. doi: 10.1002/jgrd.50305

---

## Referee Comment (RC2) · D. Gallego (Referee) · 4 Oct 2016

First, I would like to emphasize that I really like the methodology developed by Molnos et al. to characterize the jet stream. I think that the authors have developed an interesting new method able to cope with the extraordinarily complex structure of these systems. The presented results suggests that the method is able to "link" maxima in the hemispherical wind field in a way consistent with the jet stream structure, and in this sense, I think that this manuscript can be considered for publication in Earth System Dynamics. However, I have some concerns I detail in the next paragraphs:

Scientific issues:

1. Table 1 shows that after the calibration procedure, the weight $w_3$ (related to the jet latitude goes from 0.92 to 0.95. If I am correct, this implies that the cost function

essentially accounts for the local latitude, while the terms related to wind speed and direction are almost negligible. Why it is necessary to retain the terms X and Y in the cost function with such small weights? Would the resulting jet path be different if those terms are simply not considered?

2. Along the text, the authors describe some "constrains" they needed to close their algorithm without clear justification. For example (see page 4, line 5) they limit PFJs to be between 30°N-90°N latitudes and state that this is "something which does not affect the results". How sensitive is the method to changes in the 30°N threshold? Has this limit been explicitly tested? I know that is not frequent at all, but It seems to me that locally, the polar front jet (or some of its branches) could be occasionally close to the 30°N limit. Other example appears in page 6, line 9. The authors establish that they set the weight w2 "manually". How was this done? (Need clarification). Same for page 7, line 12 (and table 1).

3. In view of the examples shown in figures 5 and 6 and in the climatology (figures 11 to 14) it seems clear that the algorithm is doing quite a good work locating both jets. In this point I really miss a comparison with other similar schemes like those of Archer and Caldeira. (2008), Pena-Ortiz et al (2013) or Rikus (2015). In particular it would be very interesting a comparison related to the averages and trends of the jets (strength, mean latitude or even prevalent wave-number). Such an addition would largely improve the scientific value of this work. (I know that the new climatology represent 15-day periods, but anyway, for long term trends and averages, the new method should provide results comparable to those obtained with daily approaches).

Formal issues:

1. When the detection scheme is developed (sections 3 and 4), the authors first show the results of their first attempt (section 3), only to conclude that it did not worked (see Figure 2). Then, they start a new section (section 4), which is devoted to explain how the authors calibrated their first scheme in order to properly separate polar and

subtropical jets. Moreover, section 4 is not completely clear. In occasions the text goes "back and forth", anticipating concepts that are only developed in a following section (see for example lines 5 or 26 in page 6). Of course I know that this comment reflects mostly my personal taste, but in my opinion, section 3 and 4 could be rewritten and merged, avoiding the mention of the first not-working attempt (and thus Figure 2) and describing at once, and in a "linear" and concise way, the final working scheme.

2. Equation 1 is not consistent with the text (page 3, line 13).

3. The explanation of the Rikus' algorithm (section 4.1) is not clear. In particular the text between lines 12 and 18 could be rewritten in order to better explain the basis of this algorithm and the concepts involved (as for example how are defined the maximum (minimum) filters and the maximum (minimum) stencils).

4. Figures 11 to 14 (seasonal climatology) are very interesting because they give simple and very visual information about the new jet climatology. On the other hand, Figure 10 (annual climatology) is a little bit redundant.

5. I do not see the point in considering a single figure as a supplementary material (Figure S1). If the authors think this figure is necessary, they should include it in the main text and add some more discussion. If not, it would be better to remove it.

6. Figure 7: The caption should indicate only what is displayed by the figure, leaving the discussion for the main text.

7. Finally, the text has a number of errata (capitals, latitudes of 140°N, typos in equations, missing years in references, etc.). Please! Do a careful revision prior to publication.

Despite these comments, I would like to highlight that I consider the method quite interesting, and providing the authors perform some rewriting of the text, justify some of the thresholds they used and add some comparison with similar works, this paper is a valuable addition to the scientific literature related to the jet stream.

[Figure]

References.

Archer, C. L., and K. Caldeira (2008), Historical trends in the jet streams, Geophys. Res. Lett., 35, L08803, doi:10.1029/2008GL033614

Pena-Ortiz, C., Gallego, D., Ribera, P., Ordonez, P. and Del Carmen Alvarez-Castro, M.: Observed trends in the global jet stream characteristics during the second half of the 20th century, J. Geophys. Res. Atmos., 118, 2702–2713, doi:10.1002/jgrd.50305, 2013.

Rikus, L.: A simple climatology of westerly jet streams in global reanalysis datasets part 1 : mid latitude upper tropospheric jets, Clim. Dyn., doi:10.1007/s00382-015-2560-y, 2015.

---

## Referee Comment (RC3) · Anonymous Referee #3 · 10 Oct 2016

Recommendation: Accepted after major revisions

The authors have developed a new automatic algorithm relying on the Dijkstra's shortest path algorithm, to detect the track and meridional meandering of the PFJ and STJ on a daily basis, around the Northern Hemisphere. It comes as the natural continuation of the previous Rikus' method giving the average jet latitude.

Some aspects call for a more detailed explanation.

Main comments:

1 - The parameters w1, w2, w3 give weights in the edge cost function, respectively of the wind speed, the collinearity between the wind and edge and finally the deviation from a fixed latitude. The STJ and PFJ solutions issued from the Dijkstra's algorithm

are quite sensitive to the chosen parameter (as seen in Fig. 2 using untuned parameters). The untuned parameter values (Table 1) give the largest value to the jet latitude-guidance term. In order to provide realistic values of the jets, an educated guess of w3 (quite close to 1) is provided, coming from minimization of 6 by simulated annealing. It constrains the solution to be quite linked to the Rikus' solution. The weight w3 is probably linked to the flatness of the function x3 around the phi-clim latitude. By using a sharper function (power 8 instead of 4) weighting latitude deviations will lead to a smaller tuned w3. In fact, the optimal weights depend on the range of x1, x2 and x3 and of the particular choices of the functions x1, x2, x3 giving the weights to the edges. More possibilities exist (ex. the wind projection along the edge unitary vector could be used to substitute weights x1 and x2). Authors shall refer to the different possibilities in the method (section 2).

2- The method is not clear about the optimization of the pressure level of the jet. At which level are computed the winds entering in the method. Is it varying daily or set fixed? There is no explicit vertical guidance of the jets. How do authors deal with this aspect ?

3 – Page 8, line 10. In the discussion of Fig. 7 the algorithm does not resolve properly the PFJ and STJ. In fact, there are other not resolved topologically complex situations like when the jet splits into two branches. Authors should comment that providing hints for solving those issues.

4 - Section 4.2 about the optimization of parameters is too simplistic. A much detailed description is needed. Some points are not clear. The cost function 6 is varying with time. Therefore parameters w1, w2 and phi-clim minimizing it should also depend on time. However, the parameters are set to fixed values for the cold and for the warm season. Therefore, in order to keep consistency, the cost-function 6 should be a seasonal average. Authors should correct and clarify this point.

Technical Corrections

Pg. 1, line 30 kay → key Pg. 2, line 16 linked → are linked Pg. 2, line 21 each → each one Pg. 2, line 22-24 the sentence is rather confusing, rewrite it Pg. 2, line 27 date=2005

Fig. 9 (caption) should refer to STJ, not PFJ.

Table 1 : The start parameters does not sum 1 in agreement with 1. Please correct.

Review of the manuscript "A network-based detection scheme of the jet stream core" by Sonja Molnos, Tarek Mamdouh, Stefan Petri, Thomas Nocke, Tino Weinkauf and Dim Coumou.

Recommendation: Accepted after major revisions

The authors have developed a new automatic algorithm relying on the Dijkstra's shortest path algorithm, to detect the track and meridional meandering of the PFJ and STJ on a daily basis, around the Northern Hemisphere. It comes as the natural continuation of the previous Rikus' method giving the average jet latitude.

Some aspects call for a more detailed explanation.

Main comments:

1 - The parameters w1, w2, w3 give weights in the edge cost function, respectively of the wind speed, the collinearity between the wind and edge and finally the deviation from a fixed latitude. The STJ and PFJ solutions issued from the Dijkstra's algorithm are quite sensitive to the chosen parameter (as seen in Fig. 2 using untuned parameters). The untuned parameter values (Table 1) give the largest value to the jet latitude-guidance term. In order to provide realistic values of the jets, an educated guess of w3 (quite close to 1) is provided, coming from minimization of 6 by simulated annealing. It constrains the solution to be quite linked to the Rikus' solution. The weight w3 is probably linked to the flatness of the function x3 around the phi-clim latitude. By using a sharper function (power 8 instead of 4) weighting latitude deviations will lead to a smaller tuned w3. In fact, the optimal weights depend on the range of x1, x2 and x3

and of the particular choices of the functions x1, x2, x3 giving the weights to the edges. More possibilities exist (ex. the wind projection along the edge unitary vector could be used to substitute weights x1 and x2). Authors shall refer to the different possibilities in the method (section 2).

2- The method is not clear about the optimization of the pressure level of the jet. At which level are computed the winds entering in the method. Is it varying daily or set fixed? There is no explicit vertical guidance of the jets. How do authors deal with this aspect ?

3 – Page 8, line 10. In the discussion of Fig. 7 the algorithm does not resolve properly the PFJ and STJ. In fact, there are other not resolved topologically complex situations like when the jet splits into two branches. Authors should comment that providing hints for solving those issues.

4 - Section 4.2 about the optimization of parameters is too simplistic. A much detailed description is needed. Some points are not clear. The cost function 6 is varying with time. Therefore parameters w1, w2 and phi-clim minimizing it should also depend on time. However, the parameters are set to fixed values for the cold and for the warm season. Therefore, in order to keep consistency, the cost-function 6 should be a seasonal average. Authors should correct and clarify this point.

Technical Corrections

Pg. 1, line 30 kay → key Pg. 2, line 16 linked → are linked Pg. 2, line 21 each → each one Pg. 2, line 22-24 the sentence is rather confusing, rewrite it Pg. 2, line 27 date=2005

Fig. 9 (caption) should refer to STJ, not PFJ.

Table 1 : The start parameters does not sum 1 in agreement with 1. Please correct.

---

## Author Comment (AC1) · 26 Oct 2016

The comment was uploaded in the form of a supplement:
http://www.earth-syst-dynam-discuss.net/esd-2016-37/esd-2016-37-AC1-supplement.pdf

---

## Author Comment (AC2) · 26 Oct 2016

We are pleased with the generally positive reviewer remarks and thank the reviewer for the invested time and the very helpful comments provided, which will help us to improve the manuscript. A pointwise reply to the reviewer's comment is given below.

Specific Comments:

1.) *Table 1 shows that after the calibration procedure, the weight w3 (related to the jet latitude goes from 0.92 to 0.95. If I am correct, this implies that the cost function essentially accounts for the local latitude, while the terms related to wind speed and direction are almost negligible. Why it is necessary to retain the terms X and Y in the cost function with such small weights? Would the resulting jet path be different if those terms are simply not considered?*

It is correct that w3 (latitudinal steering parameter) is important but it is still necessary to retain the terms X and Y. With w3=1 (and without term X, which accounts for the strength of the wind field the jet stream core) would give just a straight line at $\phi_{clim}$, since this would be the minimal cost. Without the term Y the jet stream curve would be not smooth and locally spiky. For those reasons, we never tried to consider only term Z, but we did even test lower weights for X and Y, which showed us this behaviour.

2.) *Along the text, the authors describe some "constrains" they needed to close their algorithm without clear justification. For example (see page 4, line 5) they limit PFJs to be between 30◦N-90◦N latitudes and state that this is "something which does not affect the results". How sensitive is the method to changes in the 30◦N threshold? Has this limit been explicitly tested? I know that is not frequent at all, but It seems to me that locally, the polar front jet (or some of its branches) could be occasionally close to the 30◦N limit. Other example appears in page 6, line 9. The authors establish that they set the weight w2 "manually". How was this done? (Need clarification). Same for page 7, line 12 (and table 1).*

We used this latitude constraint only to speed up the code, but this is easily be changed. In fact we rerun our analyses using the full latitudinal range and get the same results with and without this constraint. For that reason, we will rewrite this part as we use the boundaries *0◦N-90◦N latitudes*.
Regarding w2: We set manually different values for w2 and looked at different plots to decide, which values gives the best results. Again this parameter is not important for the circumglobal path of the detected jet core but only for local smoothing.

We will rephrase this part in the manuscript to make it clearer.

3.) *In view of the examples shown in figures 5 and 6 and in the climatology (figures 11 to 14) it seems clear that the algorithm is doing quite a good work locating both jets. In this point I really miss a comparison with other similar schemes like those of Archer and Caldeira. (2008), Pena-Ortiz et al (2013) or Rikus (2015). In particular it would be very interesting a comparison related to the averages and trends of the jets (strength, mean latitude or even prevalent wave-number). Such an addition would largely improve the scientific value of this work. (I know that the new climatology represent 15-day periods, but anyway, for long term trends and averages, the new method should provide results comparable to those obtained with daily approaches)*

We agree with the author and will add an additional discussion section in the manuscript, where we compare strength and mean-latitudinal trends of our analysis, compare them with the literature and will provide a table of the trend analysis.
We already did such analyse and could conclude that the results are very consistent with the existing literature mentioned by reviewer.

Formal issues

We agree with the referee and rephrase as suggested.

---

## Author Comment (AC3) · 26 Oct 2016

We are pleased with the generally positive reviewer remarks and thank the reviewer for the invested time and the very helpful comments provided. These will certainly help us to improve the manuscript.

A pointwise reply to the reviewer's comment is given below.

Main Comments:

1.) *The parameters w1, w2, w3 give weights in the edge cost function, respectively of the wind speed, the collinearity between the wind and edge and finally the deviation from a fixed latitude. The STJ and PFJ solutions issued from the Dijkstra's algorithm are quite sensitive to the chosen parameter (as seen in Fig. 2 using untuned parameters). The untuned parameter values (Table 1) give the largest value to the jet latitude- guidance term. In order to provide realistic values of the jets, an educated guess of w3 (quite close to 1) is provided, coming from minimization of 6 by simulated annealing. It constrains the solution to be quite linked to the Rikus' solution. The weight w3 is probably linked to the flatness of the function x3 around the phi-clim latitude. By using a sharper function (power 8 instead of 4) weighting latitude deviations will lead to a smaller tuned w3. In fact, the optimal weights depend on the range of x1, x2 and x3 and of the particular choices of the functions x1, x2, x3 giving the weights to the edges. More possibilities exist (ex. the wind projection along the edge unitary vector could be used to substitute weights x1 and x2). Authors shall refer to the different possibilities in the method (section 2).*

   We agree with the referee that there exist different possibilities to define the cost function of an edge as for example suggested by the referee to use the wind projection along the edge unitary vector instead of condition X and Y. In addition, it is also possible to use other functions for Z (e.g. a sharper function (power 8 instead of 4)).

   We will rephrase this in the manuscript. We want to add a section to the discussion outlining limitations and possible changes and improvements to the current scheme, which might be test in future work.

   We agree that a sharper function leads to lower values of w3. The current choice still allows free movement within a latitudinal belt of roughly $\pm 20\%$ of the climatological mean and therefore the large value of $w_3$ is admissible.
   Note that in this case a sharper function means a function of lower order (e.g. quadratic), because the function is normalized by the maximum of the interval (see eq. (4)).

2.) *The method is not clear about the optimization of the pressure level of the jet. At which level are computed the winds entering in the method. Is it varying daily or set fixed? There is no explicit vertical guidance of the jets. How do authors deal with this aspect ?*

   As explained in section 2, we take a vertical average of the 3D wind velocity field resulting in a 2 dimensional field. Hence there is no height dependency, so the optimization of the pressure level of the jet is not required.
   However, in principle including the vertical dimension (3D detection scheme), and thus taking into account the pressure level , could be done in the same way as outlined in the manuscript for 2D.

3.) *Page 8, line 10. In the discussion of Fig. 7 the algorithm does not resolve properly the PFJ and STJ. In fact, there are other not resolved topologically complex situations like when the jet splits into two branches. Authors should comment that providing hints for solving those issues.*

   We would like to stress that fig.7 represents one of the difficult cases and that's also why we show it. Overall, the scheme works very well. Even in fig. 7 one can argue whether or not the jets are properly resolved.
   For example, the STJ core could split between -180 - -100° longitude, but since the wind field between 150°-180° at 40° latitude continuing at -180° longitude and 40° latitude has the stronger velocity, it is the preferable state for the STJ.

In addition, the path of the STJ over western pacific (150°-180°) is clear with very strong winds at a latitude of 40°. The path found by our algorithm over the eastern Pacific (-180° - -100°) is thus a logical extension of that across the date-line. Due to the visualization of the Pacific Ocean on opposite ends of the map in fig 7 it appears that the STJ is not properly resolved over the western Pacific, but this is rather a visual artifact.

It is important to stress that our method is *objective* and hence there are cases the algorithm finds a path, which differs from the path, which one would assume by visual choice.

To account for splitting of the STJ and PFJ, the easiest way would be to calculate not 2 but 4 (or even more) jet stream cores with different climatological jet stream latitudes phi-clim. In cases, where only one path exists, the found jet stream cores would be combined to one path and in other cases, where two paths exist, they would split. We will add this to the discussion section for possible future improvements.

4.) *Section 4.2 about the optimization of parameters is too simplistic. A much detailed description is needed. Some points are not clear. The cost function 6 is varying with time. Therefore parameters w1, w2 and phi-clim minimizing it should also depend on time. However, the parameters are set to fixed values for the cold and for the warm season. Therefore, in order to keep consistency, the cost-function 6 should be a seasonal average. Authors should correct and clarify this point.*

w1, w2 and phi-clim are independent in time and change only for the warm and cold season. There was a typo in eq. (6), the skill function is the sum of all time steps in warm season or cold season:

$$S = \sum_{t=1}^{t_{end}} \sqrt{(\phi_{Rikus}(t) - \phi_{mean}(t))^2},$$

( 6 )

whereby $\phi_{mean}(t)$ is the zonal mean latitude calculated with our algorithm, $\phi_{Rikus}(t)$ is the jet stream core determined by Rikus' algorithm for time period $t$ and $t_{end}$ is the number of 15-day running mean time step, where Rikus' algorithm finds a jet core.

Technical corrections

We agree with the referee and will rephrase as suggested.

---

## Author Comment (AC4) · 26 Oct 2016

We are pleased with the generally positive reviewer remarks and thank the reviewer for the invested time and the very helpful comments provided, which will help us to improve the manuscript. A pointwise reply to the reviewer's comment is given below.

Specific Comments:

1.) *I'm not so sure that algorithms to detect jet cores are lacking (as stated in the abstract). There are actually a relatively large number of previously published papers which are based on either a single level/layer or zonal/sectorial mean latitude-pressure fields. The current application of the network-based method is yet another variation of this and so needs to be put in context with other (similar) methods which use single level or mean-layer wind fields (e.g. Koch et al, 2006, Archer & Caldiera 2008, Pena- Ortiz et al 2013). Hence at the risk of expanding the paper too much I think it would be useful to acknowledge some more of the previous work and to compare with the results obtained here (even though they are based on a 15 day mean as opposed to monthly means) and discuss why this method has advantages over the previous studies.*

We will rephrase as suggested.

2.) *I think 'time step' is a confusing choice of phrase to describe the 15 day means – maybe use 'time period' ?*

We will rephrase as suggested.

3.) *The simulated annealing actually uses the Rikus algorithm so it is being used as more than just a comparison. (page 1, line 18) and the abstract description should reflect that.*

We will rephrase as suggested.

4.) *Were the original runs (Fig. 2) done with with the un-optimised weights from table 1? If not what was used?*

The original runs were done with un-optimised weights from table 1. We will rephrase this part to make it clearer.

5.) *The supplementary plot (S1) is only mentioned in a single sentence without sufficient context to make it worth while. Either add more discussion or remove it?*

With this plot, we would like to show that our method is able to track also omega-shape pattern, but in principle we agree, it is not necessary in order to present our algorithm and we will remove it.

Technical Corrections.

In general, we agree with the referee and we answer only specific questions raised by the reviewer:

**Abstract Line 21:** In this case latitudes (and not longitudes) were meant: We present probabilistic, regionally distinct positions for both jets for all seasons. This shows that winter is characterized by two well separated jets at mean latitudes of 20°S and 140°N.

**Page 4, lines1,2:** This means that the path of the jet stream core is not an injective function. This way, omega-shaped jet stream paths are possible.

**Page 4, lines 12-13:** The sum of the weights is 1.

**Page 7,lines 7-8:** Since Rikus' algorithm finds a subtropical jet for almost all time steps, we used at first every 14[th] of the found subtropical jet stream core for optimization.

**Page 8, line 20:** We agree with the referee: These differences are due to the undulations explained above.

---

## Author Response (AR1)

Dear Editor,

Herewith we submit our revised manuscript entitled "A network-based detection scheme for the jet stream core" by Sonja Molnos et al..

We were very pleased to read the constructive comments of the three reviewers and we appreciated their suggestions which have improved our manuscript both in terms of readability and content.

We feel that these changes have greatly improved our manuscript. We have dealt with all reviewer comments and below you will find a point-by-point response to all of them.

We very much look forward to hearing from you. Yours sincerely,

Sonja Molnos (on behalf of all authors)

**Point by point response to reviewer comments (comments in italic and our response indented).**

**Reviewer 1**

I think a minimalist change to the title would probably be "A network-based detection scheme for the jet stream core".

We agree with the reviewer and changed the title accordingly.

**Specific Comments:**

**1**. I'm not so sure that algorithms to detect jet cores are lacking (as stated in the abstract). There are actually a relatively large number of previously published papers which are based on either a single level/layer or zonal/sectorial mean latitude-pressure fields. The current application of the network-based method is yet another variation of this and so needs to be put in context with other (similar) methods which use single level or mean-layer wind fields (e.g. Koch et al, 2006, Archer & Caldiera 2008, Pena- Ortiz et al 2013). Hence at the risk of expanding the paper too much I think it would be useful to acknowledge some more of the previous work and to compare with the results obtained here (even though they are based on a 15 day mean as opposed to monthly means) and discuss why this method has advantages over the previous studies.

We rewrote the abstract accordingly (p. 1, l. 12 - 14):

Some algorithms exist which can detect the 2D (latitude and longitude) jets' core around the hemisphere, but all of them use a minimal threshold to determine the subtropical and polar jet stream. This is particularly problematic for the polar jet stream whose wind velocities can change rapidly from very weak to very high values and vice versa.

We rephrased as suggested and added the following text in the introduction part (p. 2, l. 29 - p.3, l. 18):

A method for calculating the jet stream core in latitude-longitude-direction was developed by Archer and Caldeira (2008). They define the jet's latitudinal position for each longitude using mass-flux weighted monthly mean wind speed between 100 and 400hPa in the northern (15N - 70N) and southern hemispheres (SHJ:40S - 15S; SHP: 70S - 40S).

Their algorithm detects only one jet position in the northern hemisphere and thus cannot distinguish between polar and subtropical jet streams. It is also not possible to capture omega-shaped jet patterns, since that method assigns only one latitude for each longitude.

Koch et al.(2006) classify so-called deep or shallow jet stream events. Their three-step algorithm first calculates the vertically averaged horizontal wind speed between two pressure levels (p1 = 100 hPa and p2 = 400 hPa) for each time instance and grid point. Next, a threshold of 30 ms-1 is applied to detect a so-called jet event in a grid cell. Further analysis over vertical layers classifies events into deep or shallow jet stream events but it does neither extract the actual stream core, nor does it distinguish between polar and subtropical jet stream (Koch et al., 2006).

Gallego et al. developed a scheme using a geostrophic streamline of maximum daily averaged velocity at 200 hPa to find the jet stream in the southern hemisphere. It uses wind velocities threshold of  $30 \ m \ s^{-1}$  and distinguishes between the subtropical and polarjet stream, when the average latitudinal difference is greater than 15°. The threshold was set by manual optimization(Gallego et al., 2005). This approach might work reasonably for the southern hemisphere jets, a fixed threshold approach is particular problematic for the northern hemisphere polar jet which can change drastically in strength on weekly timescales.

The first 3D method (longitude, latitude, height) developed by Limbach at al. (2012), detects and tracks specific properties of atmospheric features as merging and splitting jet streams (via clustering of data points). Still this method cannot distinguish between subtropical and polar jet stream and also requires the use of a wind velocity threshold (Limbach et al., 2012).

**2.** *I* think 'time step' is a confusing choice of phrase to describe the 15 day means – maybe use 'time period' ? We rephrased as suggested.

**3.** The simulated annealing actually uses the Rikus algorithm so it is being used as more than just a comparison. (page 1, line 18) and the abstract description should reflect that. We rephrased as suggested and thus added the following paragraph in the abstract (p. 1, I. 19- 21):

The parameter values of the detection scheme are optimized using simulated annealing and a skill function that accounts for the zonal-mean jet stream position (Rikus, 2015). After the successful optimization process we apply our scheme to reanalysis data covering 1979 - 2015 and calculate seasonal-mean probabilistic maps and trends in wind strength and position of jet streams.

**4.** Were the original runs (Fig. 2) done with with the un-optimised weights from table 1? If not what was used? The original runs were done with un-optimised weights from table 1. To clarify this weadded the following paragraph (p. 8, I. 14-15):

Improvements in the detected jet stream core positions due to the optimization process relative to the positions found by the untuned algorithm (Fig. 4, parameters are given Table 1) are illustrated in Fig. 5.

**5.** The supplementary plot (S1) is only mentioned in a single sentence without sufficient context to make it worth while. Either add more discussion or remove it?

With this plot, we would like to show that our method is able to track also omega-shape pattern, but in principle we agree, it is not necessary in order to present our algorithm and we removed it.

Technical Corrections.

**Abstract Line 21**: 'mean longitudes of 20S and 140N' ?????? I don't know what this is supposed to mean! In this case we meant 20°W to 140°E. We rewrote the abstract (p. 1, I. 23-24):

Page 2, line 27: No year given for Gallego et al (and in reference list). Try 2005.
We added the year for Gallego et al. (p. 3, 1.13).
Page 2, line 32: there is no reference for Limbach et al (2012).
We added the reference (p. 15, 1.24-25).
Page 3, line 2: "zonally" should be "zonal"
We rewrote as suggested (p. 3, 1.19)
Page 3, line 4: "such approach" should be "such an approach"
We rewrote as suggested (p. 3, 1. 21)
Page 3, line 13: "all different" should be "different"
We rewrote as suggested (p. 3, 1. 31)
Page 3, line 18: "for 4" should be "four"

We rewrote as suggested (p. 4, I. 4)

Page 4, lines1,2: I'm not sure what this line actually means!

This means that the path of the jet stream core is not an injective function. This way, omega-shaped jet stream paths are possible. To clarify that part we added the following paragraph (p. 4, l. 16-17):

The path itself is not an injective function of longitude meaning that the path can pass multiple times the same longitudinal coordinates.

**Page 4, lines 3,4:** "To avoid noise and reduce computational costs only those grid points where the wind velocity is greater than 10% of the maximum wind velocity for the considered time step are connected." We rewrote as suggested (p. 4, I. 18-19).

**Page 4, lines 12-13:** The description of the weights is inconsistent – is their sum 1 or

less than 1?

The sum of the weights is 1. We corrected this part (p. 5, l. 1).

Page 5, equation 4: there appear to be some brackets missing in the denominator.

We added the missing brackets (p. 5, l.18).

Page 5, line 19: "near of 65" should be "near to 65"

Due to rewriting the method part (suggested by reviewer 2), the sentence was removed.

**Page 6, line 15:** "First a maximum (minimum) filter" should be "First a local maximum (minimum) filter"

We rewrote as suggested (p. 6, l. 19-20).

**Page 7, line 5:** "With the found zonal mean subtropical and polar jet stream latitudes by Rikus" should be "With the zonal mean subtropical and polar jet stream latitudes found by Rikus' algorithm"

We rewrote as suggested (p. 7, l. 27-29).

Page 7, lines 7-8: This sentence needs to be clarified.

We rewrote the sentence in the following way (p. 8, l. 1 – 4):

For computational reasons, we first optimize the STJ parameters using every 14th time period. This first step gives us proper starting conditions for the final optimization. Thus, in the final optimization we include all time periods and used as starting point the optimized parameters found in the first step, which strongly speeds up convergence of the annealing method.

Page 7, line21: "it is more undulated" should be "it undulates more"

We rephrased as suggested (p. 8, I.14-15).

Page 7, lines 22-23: Try - "Improvements in the jet stream core positions due to the

optimization process relative to the positions found by the untuned ..."

We rephrased as suggested (p. 8, l. 16-17).

Page 8, line 7: "polar Jet" should be "polar jet"

We rewrote as suggested (p. 8, I. 28)

Page 8, line 16: "not between minimum and maximum latitude" should be "not between

the minimum and maximum latitude"

We rephrased as suggested (p. 9, l. 11-12).

Page 8, line 17: "equivalent of 6.4" should be "equivalent to 6.4"

We rewrote as suggested (p. 9, l. 12).

Page 8, line 20: Change to "These differences are due to the undulations explained

above." ?

We rewrote as suggested (p. 9, l. 15).

**Page 9, lines 16-34:** The language needs to be cleaned up – the section does not scan well with a number of missing 'the' and 'a's.

We cleaned the language up and rephrased the paragraph to (p. 10, l. 7-26):

This coexistence of the STJ and PFJ in the eastern hemisphere, compared to more frequent merged jet states in the western hemisphere, is well documented in the literature, but was never shown in probabilistic plots as presented here (Eichelberger and Hartmann, 2007; Li and Wettstein, 2012; Son and Lee, 2005; Woollings, 2010). Those different jet stream states occur, since the processes which lead to their existence operate and interact in non-linear ways (Harnik et al., 2016; Lee and Kim, 2003). In the North Atlantic, STJ and PFJ are separated because the region of strongest baroclinicity is located relatively far poleward. In contrast, the region of strongest baroclinicity in the North Pacific islocated near the latitude of maximum zonal wind, favouring a merged jet (Lee and Kim, 2003; Li and Wettstein, 2012). Such a merged jet stream is also called the eddy-thermally driven jet because of the two different genesis mechanisms. In special cases, there is the possibility that this eddy-thermally driven jet stream also appears over the North Atlantic (Harnik et al., 2014). This happens if the tropical forcing strengthens or the mid-latitude baroclinicity weakens.

In addition, the panels (b) give probabilities of the zonal-mean latitude of both jets, showing enhanced variability of the PFJ compared to the STJ. The range of overlapping latitudes between STJ and PFJ is larger in summer than in winter because of the poleward shift of the STJ. The latitudinal variability in STJ is lower in summer and winter than in spring and autumn, whereas the variability of the PFJ is similar between seasons. However, the location of the maximum in the PFJ histogram changes per season: in winter, the maximum is at ca 55°N, whereas in summer there are two maxima at 50°N and at 70°N. These two maxima probably reflect the different behaviour in western and eastern hemisphere in the PFJ. In spring, there is no clear maximum visible (between 40°N-60°N), and in autumn it is again close to 55°N.

To quantify those merged and separated states further, one could use the latitudinal difference between STJ and PFJ, for all longitudes, and this way create the probability density distributions of merged and separated jets. The presented results (Fig. 10 - 13) might in principle also be the result of clearly separated jets which displace latitudinally over time to create the overlapping probability density.

Fig. 4 caption (page 15, line 14): the points in (f) are blue not white.
We changed the text as suggested (Fig. 3, p. 18, I.8).
Fig. 5 caption (page 16, line 4): should be "compare with Fig. 2"
We rewrote as suggested (Fig. 5, p. 19, I.9)
Fig. 6 caption (page 16, line 7): Remove ", compare Fig. 2." - it is not necessary.
We removed as suggested (Fig. 6, p. 20, I.3).

**Reviewer 2**

**1.** Table 1 shows that after the calibration procedure, the weight w3 (related to the jet latitude goes from 0.92 to 0.95. If I am correct, this implies that the cost function essentially accounts for the local latitude, while the terms related to wind speed and direction are almost negligible. Why it is necessary to retain the terms X and Y in the cost function with such small weights? Would the resulting jet path be different if those terms are simply not considered?

It is correct that  $w_3$  (latitudinal steering parameter) is important but it is still necessary to retain the terms X and Y. With  $w_3 = 1$  (and without term X, which accounts for the strength of the wind field of the jet stream core) the algorithm would give just a straight line at  $\phi_{clim}$ , since this would be the minimal cost. Without the term Y the jet stream curve would be not smooth and locally spiky. For those reasons, we never tried to consider only term Z, but we did even test lower weights for X and Y, which showed us this behaviour. We added a paragraph to clarify this part (p. 8, l. 11-12):

We would like to emphasize that all terms are important even though  $w_3$  has the biggest value. If we would consider only  $Z_j$ , and exclude all other terms, the jet stream core would be a straight line at  $\phi_{\text{clim}}$ , since this would be the shortest path.

**2.** Along the text, the authors describe some "constrains" they needed to close their algorithm without clear justification. For example (see page 4, line 5) they limit PFJs to be between 30°N-90°N latitudes and state that this is "something which does not affect the results". How sensitive is the method to changes in the 30°N threshold? Has this limit been explicitly tested? I know that is not frequent at all, but It seems to me that locally, the polar front jet (or some of its branches) could be occasionally close to the 30°N limit. Other example appears in page 6, line 9. The authors establish that they set the weight w2 "manually". How was this done? (Need clarification). Same for page 7, line 12 (and table 1).

We used this latitude constraint only to speed up the code, but this is easily be changed. In fact we rerun our analyses using the full latitudinal range and get the same results with and without this constraint. For that reason, we rephrased this part in the manuscript (p. 4, l. 20-22) as we use the boundaries  $O \circ N-9O \circ N$  latitudes: In order to reduce computational costs, the spatial domain is reduced to the main region of interest  $O^\circ -75^\circ N$  for the subtropical jet stream on the northern hemisphere. The spatial domain for the polar jet stream is  $O^\circ N-90^\circ N$ , since in some rare cases the polar jet stream could be occasionally close to the  $30^\circ N$  limit.

Regarding  $w_2$ : We set manually different values for w2 and looked at different plots to decide which values gives the best results. Again this parameter is not important for the circumglobal path of the detected jet core but only for local smoothing.

We added a paragraph in p.7, l. 21-23.:

For the manual tuning of  $w_2$ , we tried different values for different time periods and found a value of 0.0015 to give the most desirable results. Since this weighting factor only affects local smoothing, its value does not affect the hemispheric path found.

**3.** In view of the examples shown in figures 5 and 6 and in the climatology (figures 11 to 14) it seems clear that the algorithm is doing quite a good work locating both jets. In this point I really miss a comparison with other similar schemes like those of Archer and Caldeira. (2008), Pena-Ortiz et al (2013) or Rikus (2015). In particular it would be very interesting a comparison related to the averages and trends of the jets (strength, mean latitude or even prevalent wave-number). Such an addition would largely improve the scientific value of this work. (I know that the new climatology represent 15-day periods, but anyway, for long term trends and averages, the new method should provide results comparable to those obtained with daily approaches)

We added an additional section in the manuscript (as well as an additional paragraph in the abstract and the summary), where we compared strength and mean-latitudinal trends of our analysis, compared them with the literature and provided a table of the trend analysis (p. 1, l. 20-22 & p. 11, l. 2-29 & p. 12, l. 28-30, l. 1 & table 2 & Fig. 14):

**5 Global trends**

Fig 14 shows trends in the latitudinal position and wind velocity for summer and winter as well as annual data

derived from our Dijkstra –Jet-detection scheme. Table 2 summarizes the results giving linear trends in mean Jet stream latitude and mean wind velocity with bold values indicating statistical significance (p<0.05).

In order to compare our results with literature results, we calculated mean Jet stream latitude and mean wind velocity trends, which are shown in Table 2. Bold values indicate statistical significance (p<0.05). We used Monte Carlo analysis with 10000 surrogate time series of shuffled data to determine significance (Di Capua and Coumou, 2016; Pollard and Lakhani, 1987; Schreiber and Schmitz, 2000). To account for the fact that running means present not truly independent data, we shuffle blocks of 15 days in this method.

In general, we observe a northward trend for the STJ (except for SON) which is significant for winter and annual time series. The latitudinal position of the PFJ shows more mixed behavior with different signs for different seasons. A pronounced and significant equatorward trend is detected for the PFJ in winter. Wind velocities have generally weakened for both STJ and PFJ, something which is significant for summer, in agreement with Coumou et al. (2015) and Lehmann and Coumou (2015).

Overall these reported trends are in good agreement with previous studies though it is somewhat difficult to make direct comparisons as different studies analyzed different aspects of the flow field. For example, Pena-Ortiz et al. (2013) did not calculate separate trends for the STJ and PFJ, but instead for different range of latitudes: for winter 15°-40°, for spring and autumn 10°-70° as well as for summer 30°-60°. Since STJ winds are in general stronger, we assume that at least for spring, summer and autumn their reported trends reflect trends of the STJ. Similarly, Archer & Caldeira (2008) considered only trends in NH jet stream between 15°N-70°N, where we again expect that this mostly reflects the behavior of the STJ. Rikus (2015) calculated trends for one northern jet stream core within 20°N-54°N, so we can assume that the trend most probably describes the trend of the STJ. The findings of those studies can thus be best compared to our STJ findings. The annual poleward trend in latitudinal position of the STJ, detected with our method, is consistent with the results of Rikus and Archer & Caldeira. Also the latitudinal trend in summer calculated by our method has the same sign and order of magnitude as in Rikus and Pena-Ortiz et al, but the trend in winter is greater in our and Rikus' method compared to that from Pena-Ortiz. The trends for spring and autumn agree in sign with the analysis of Pena-Ortiz using 20th century data, but are weaker and even change sign for the NCEP/NCAR data set in autumn.

The wind velocity trends are positive in the publication of Pena-Ortiz, whereas we observed a negative trend as Rikus (except summer) and Archer & Caldeira. With our more-advanced approach which is able to differentiate between subtropical and polar jet, we detect stronger (and mostly significant) weakening compared to the other studies.

**Formal Issues:**

**1.** When the detection scheme is developed (sections 3 and 4), the authors first show the results of their first attempt (section 3), only to conclude that it did not worked (see Figure 2). Then, they start a new section (section 4), which is devoted to explain how the authors calibrated their first scheme in order to properly separate polar and subtropical jets. Moreover, section 4 is not completely clear. In occasions the text goes "back and forth", anticipating concepts that are only developed in a following section (see for example lines 5 or 26 in page 6). Of course I know that this comment reflects mostly my personal taste, but in my opinion, section 3 and 4 could be rewritten and merged, avoiding the mention of the first not-working attempt (and thus Figure 2) and describing at once, and in a "linear" and concise way, the final working scheme.

We rewrote section 3 and 4 as suggested and merged them to the section "3 methods" (p. 4, 1.8 - p.9, 1.19) and changed also the outline accordingly (p. 3, 1.29 - 32).

**2.** Equation 1 is not consistent with the text (page 3, line 13). The sum of the weights is 1. We corrected this part (p. 5, l. 1).

**3.** The explanation of the Rikus' algorithm (section 4.1) is not clear. In particular the text between lines 12 and 18 could be rewritten in order to better explain the basis of this algorithm and the concepts involved (as for example how are defined the maximum (minimum) filters and the maximum (minimum) stencils). We rewrote the explanation of Rikus' part for clarity in p. 6, I. 19-25:

Figure 3 shows the scheme of Rikus' algorithm. First a local maximum (minimum) filter is applied to the original zonal mean U field. The maximum (minimum) filter is defined as a 25 point maximum stencil (25 point minimum stencil) applied to the total U field. The stencil algorithm replaces the maximum (minimum) value within a box of 5 points in x- and y- direction (resulting in a total 25 grid points) to the central grid point of that box. The box with the central grid point (x, y) moves over the total U field starting at the upper left corner of the zonal mean U field and ending at the at the lower right corner.

This way the fields  $U_{\rm Min}$  and  $U_{\rm Max}$  are determined (Fig. 3 b, c).

**4.** Figures 11 to 14 (seasonal climatology) are very interesting because they give simple and very visual information about the new jet climatology. On the other hand, Figure 10 (annual climatology) is a little bit redundant. We agree with the reviewer and removed Figure 10.

**5.** I do not see the point in considering a single figure as a supplementary material (Figure S1). If the authors think this figure is necessary, they should include it in the main text and add some more discussion. If not, it would be better to remove it.

As explained above, with this plot, we would like to show that our method is able to track also omega-shape pattern, but in principle we agree, it is not necessary in order to present our algorithm and we will remove it.

**6.** Figure 7: The caption should indicate only what is displayed by the figure, leaving the discussion for the main text. We removed the discussion from the caption of Figure 7 as suggested.

**7.** Finally, the text has a number of errata (capitals, latitudes of 140N, typos in equations, missing years in references, etc.). Please! Do a careful revision prior to publication. We corrected all errata.

**Reviewer 3**

**1.** The parameters w1, w2, w3 give weights in the edge cost function, respectively of the wind speed, the collinearity between the wind and edge and finally the deviation from a fixed latitude. The STJ and PFJ solutions issued from the Dijkstra's algorithm are quite sensitive to the chosen parameter (as seen in Fig. 2 using untuned parameters). The untuned parameter values (Table 1) give the largest value to the jet latitude- guidance term. In order to provide realistic values of the jets, an educated guess of w3 (quite close to 1) is provided, coming from minimization of 6 by simulated annealing. It constrains the solution to be quite linked to the Rikus' solution. The weight w3 is probably linked to the flatness of the function x3 around the phi-clim latitude. By using a sharper function (power 8 instead of 4) weighting latitude deviations will lead to a smaller tuned w3. In fact, the optimal weights depend on the range of x1, x2 and x3 and of the particular choices of the functions x1, x2, x3 giving the weights to the edges. More possibilities exist (ex. the wind projection along the edge unitary vector could be used to substitute weights x1 and x2). Authors shall refer to the different possibilities in the method (section 2).

We agree with the referee that there exist different possibilities to define the cost function of an edge as for example suggested by the referee to use the wind projection along the edge unitary vector instead of condition X and Y. In addition, it is also possible to use other functions for Z (e.g. a sharper function (power 8 instead of 4)). We rephrased this in the manuscript and added to the method section (p. 6, l. 4 - 9) as well as to the discussion outlining limitations and possible changes and improvements to the current scheme, which might be test in future work (p. 12, l. 10- 19).

**3 Methods**

There are of course other slightly different ways to define wind strength, wind direction and latitudinal dependence for the edges of the network. For example,  $X_j$  and  $Y_j$  could be merged to a term, which considers the wind projection along the edge unitary vector. In addition, it is possible to use a lower- or higher ordered function for equation 4, e.g. a linear function or a function with the order of 8. However, a lower order means less free movement within the latitudinal belt centered around  $\phi_{clim}$ . A higher order has negligible effects since equation 4 within the central latitudinal belt already gives values close to zero.

**6** Summary and Discussion**

Instead of using the wind direction and wind strength, it is also possible to merge both condition and consider only the wind projection along the edge unitary vector. However, with two terms we have more flexibility regarding the weights of the terms.

In addition, the jet stream latitudinal guidance term, which is in our case a fourth order function of latitude, could be a lower- or higher ordered function like a linear function or a function with the order of 8. A lower order means less freedom for the path to move away from the climatological latitude, whereas a higher order has only little effect, since the cost of a fourth order function are already small in the latitudinal belt.

We agree that a sharper function leads to lower values of  $w_3$ . The current choice still allows free movement within a latitudinal belt of roughly ±20% of the climatological mean and therefore the large value of  $w_3$  is admissible.

Note that in this case a sharper function means a function of lower order (e.g. quadratic), because the function is normalized by the maximum of the interval (see eq. (4)).

**2.** The method is not clear about the optimization of the pressure level of the jet. At which level are computed the winds entering in the method. Is it varying daily or set fixed? There is no explicit vertical guidance of the jets. How do authors deal with this aspect?

As explained in section 2, we take a vertical average of the 3D wind velocity field resulting in a 2 dimensional field. Hence there is no height dependency, so the optimization of the pressure level of the jet is not required. However, in principle including the vertical dimension (3D detection scheme), and thus taking into account the pressure level, could be done in the same way as outlined in the manuscript for 2D. We explained that in the discussion (p. 13, l. 1-3):

Parameters for the third dimension could be optimized in the similar way as done for latitude, but using pressure heights.

**3.** Page 8, line 10. In the discussion of Fig. 7 the algorithm does not resolve properly the PFJ and STJ. In fact, there are other not resolved topologically complex situations like when the jet splits into two branches. Authors should comment that providing hints for solving those issues.

We would like to stress that fig.7 represents one of the difficult cases and that's also why we show it. Overall, the scheme works very well. Even in fig. 7 one can argue whether or not the jets are properly resolved. For example, the STJ core could split between -180 - -100° longitude, but since the wind field between 150°-180° at 40° latitude continuing at -180° longitude and 40° latitude has the stronger velocity, it is the preferable state for the STJ.

In addition, the path of the STJ over western pacific (150°-180°) is clear with very strong winds at a latitude of 40°. The path found by our algorithm over the eastern Pacific (-180° - -100°) is thus a logical extension of that across the date-line. Due to the visualization of the Pacific Ocean on opposite ends of the map in fig 7 it appears that the STJ is not properly resolved over the western Pacific, but this is rather a visual artifact.

It is important to stress that our method is *objective* and hence there are cases the algorithm finds a path, which differs from the path, which one would assume by visual choice.

To account for splitting of the STJ and PFJ, the easiest way would be to calculate not 2 but 4 (or even more) jet stream cores with different climatological jet stream latitudes phi-clim. In cases, where only one path exists, the found jet stream cores would be combined to one path and in other cases, where two paths exist, they would split. We added this to the part "3.2 Results of the optimization process" (p. 9, I. 6-7) and in the discussion section for possible future improvements (p. 13, I.3 - 5):

**3.2 Results of the optimization process**

Figure 7 shows a situation, where also other paths for the STJ and the PFJ could be considered, the jets split into two jet stream cores.

**6** Summary and Discussion**

In addition, to account for splitting of the STJ and PFJ, we plan to calculate not two, but four (or even more) jet

stream cores with different climatological mean latitude  $\phi_{clim}$ . In cases, where only one path exists, the found jet stream cores would be combined to one path (based on their similarities to each other) and in other cases, where two paths exist, they would split.

**4.** Section 4.2 about the optimization of parameters is too simplistic. A much detailed description is needed. Some points are not clear. The cost function 6 is varying with time. Therefore parameters w1, w2 and phi-clim minimizing it should also depend on time. However, the parameters are set to fixed values for the cold and for the warm season. Therefore, in order to keep consistency, the cost-function 6 should be a seasonal average. Authors should correct and clarify this point.

 $w_1$ ,  $w_2$  and  $\phi_{\text{clim}}$  are independent in time and change only for the warm and cold season. There was a typo in eq. (6), the skill function is the sum of all time steps in warm season or cold season:

$$S = \sum_{t=1}^{t_{\text{end}}} \sqrt{(\phi_{\text{Rikus}}(t) - \phi_{\text{mean}}(t))^2},$$
(6)

where  $\phi_{\text{mean}}(t)$  is the zonal mean of all latitudes found by our algorithm,  $\phi_{\text{Rikus}}(t)$  is the zonal mean latitude of the jet stream core determined by Rikus' algorithm. We take the sum of the differences in latitude for all time periods t, where Rikus'algorithm finds a jet core ( $t_{\text{end}}$  is the number of such time periods).

Technical corrections Pg. 1, line 30 kay  $\rightarrow$  key We rewrote as suggested (p. 2, l. 1). Pg. 2, line 16 linked  $\rightarrow$  are linked We rewrote as suggested (p. 2, l. 19). Pg. 2, line 21 each  $\rightarrow$  each one We rewrote as suggested (p. 2, l. 24). Pg. 2, line 22-24 the sentence is rather confusing, rewrite it We rewrote as suggested (p. 2, l. 26-27). Pg. 2, line 27 date=2005 We rewrote as suggested (p. 3, l. 13). Fig. 9 (caption) should refer to STJ, not PFJ. We rewrote as suggested (p. 21, l. 7). Table 1 : The start parameters does not sum 1 in agreement with 1. Please correct. We rewrote as suggested (p. 5, l. 1). List of changes in the main manuscript "A network-based detection scheme of the jet stream core"

p. 1., I.1: changed title from "A network-based detection scheme of the jet stream core" to "A network-based detection scheme for the jet stream core"

p. 1, I. 12-14 changed to

"Some algorithms exist which can detect the 2D (latitude and longitude) jets' core around the hemisphere, but all of them use a minimal threshold to determine the subtropical and polar jet stream. This is particularly problematic for the polar jet stream whose wind velocities can change rapidly from very weak to very high values and vice versa."

**p. 1, I. 19-24 changed to**

"The parameter values of the detection scheme are optimized using simulated annealing and a skill function that accounts for the zonal-mean jet stream position (Rikus, 2015). After the successful optimization process we apply our scheme to reanalysis data covering 1979 - 2015 and calculate seasonal-mean probabilistic maps and trends in wind strength and position of jet streams.

We present longitudinally defined probability distributions of the positions for both jets on the northern hemisphere for all seasons. This shows that winter is characterized by two well separated jets over Europe and Asia (ca. 20°W to 140°E)."

p.1, I.25 changed "and" to "but can have"

p. 1, l. 27 - 28: changed to: "With this algorithm it is possible to investigate the position of the jets' cores around the hemisphere and it is therefore well suitable to analyse jet stream patterns in observations and models, enabling more-advanced model validation."

p. 2, I. 2: changed "kay" to "key"

p. 2, I. 16 changed "could" to "can"

p. 2, l. 19: changed "linked" to "are linked"

p. 2, l. 19: added reference (Kornhuber et al., 2016)

p. 2, l. 23, changed to "Hence, jet streams play a key role in the general circulation and for generating midlatitude weather conditions and extremes."

[revised manuscript text omitted]

p. 5, l. 1: changed "their sum is less than or equal to one." to "their sum is equal to one."

p. 5, l.19: removed sentence "The detected PFJ path is very similar to the STJ path, which is explained by strong subtropical wind speeds. As clearly seen in the right panel of Fig. 2, the PFJ core is actually located near of 65°N."

p. 5, l. 19: changed

$$Z_{j} = \frac{(\phi_{j} - \phi_{\text{clim}})^{4}}{\max(\phi_{\text{clim}}, 90 - \phi_{\text{clim}})^{4}} \text{ to}$$

$$Z_{j} = \frac{(\phi_{j} - \phi_{\text{clim}})^{4}}{(\max(\phi_{\text{clim}}, 90 - \phi_{\text{clim}}))^{4}}$$

p. 6, l. 4 – 9: added: "There are of course other slightly different ways to define wind strength, wind direction and latitudinal dependence for the edges of the network. For example,  $X_j$  and  $Y_j$  could be merged to a term, which considers the wind projection along the edge unitary vector. In addition, it is possible to use a lower- or higher ordered function for equation 4, e.g. a linear function or a function with the order of 8. However, a lower order means less free movement within the latitudinal belt centered around  $\phi_{clim}$ . A higher order has negligible effects since equation 4 within the central latitudinal belt already gives values close to zero."

p. 6, l. 13: added section "**3.1 Calibration of weights**" and changed the general order of the text (first present Rikus's algorithm and then simulated annealing)

p. 6, .1. 19 - 25 changed to "Figure 3 shows the scheme of Rikus' algorithm. First a local maximum (minimum) filter is applied to the original zonal mean U field. The maximum (minimum) filter is defined as a 25 point maximum stencil (25 point minimum stencil) applied to the total U field. The stencil algorithm replaces the maximum (minimum) value within a box of 5 points in x- and y- direction (resulting in a total 25 grid points) to the central grid point of that box. The box with the central grid point (x, y) moves over the total U field starting at the upper left corner of the zonal mean U field and ending at the at the lower right corner.

This way the fields  $U_{\rm Min}$  and  $U_{\rm Max}$  are determined (Fig. 3 b, c). "

**p. 7, l. 7 – 15: changed to**

[revised manuscript text omitted]

p. 9, I. 11: changed "not between minimum and maximum latitude" to "not between the minimum and maximum latitude"

p. 9, l. 12: changed "Polar jet" to "polar jet"

p. 9, I. 12: changed "equivalent" to "equivalent to"

p. 9, l. 15 changed to "These differences are due to the undulations explained above."

p. 9, l. 22-23: changed to: "Figures 10, 11, 12 and 13 show probabilistic jet stream positions for different seasons with brown dashed contour lines representing the subtropical jet and black solid contour lines representing the polar jet."

p. 9, l. 25-29: changed to "Moreover, in summer the probability that the jets merge in the western hemisphere is higher, whereas in winter the probability that they are clearly separated over almost all longitudes is higher.

In addition, the probability frequency of the PFJ is much broader than the probability of the STJ and no clear latitudinal shift between seasons is observed. In particular in summer the PFJ distribution is smeared out (indicating large fluctuations in its position) whereas in winter it is more confined."

p. 10, l. 8 – 28: changed to "This coexistence of the STJ and PFJ in the eastern hemisphere, compared to more frequent merged jet states in the western hemisphere, is well documented in the literature, but was never shown in probabilistic plots as presented here (Eichelberger and Hartmann, 2007; Li and Wettstein, 2012; Son and Lee, 2005; Woollings, 2010). Those different jet stream states occur, since the processes which lead to their existence operate and interact in non-linear ways (Harnik et al., 2016; Lee and Kim, 2003). In the North Atlantic, STJ and PFJ are separated because the region of strongest baroclinicity is located relatively far poleward. In contrast, the region of strongest baroclinicity in the North Pacific is located near the latitude of maximum zonal wind, favouring a merged jet (Lee and Kim, 2003; Li and Wettstein, 2012). Such a merged jet stream is also called the eddy-thermally driven jet because of the two different genesis mechanisms. In special cases, there is the possibility that this eddy-thermally driven jet stream also appears over the North Atlantic (Harnik et al., 2014). This happens if the tropical forcing strengthens or the mid-latitude baroclinicity weakens.

[revised manuscript text omitted]

p. 12, I. 20 added "and trends"

p. 12, l. 20: added "with prior work"

p. 12, l. 28 – 31: added "We reported trends of the mean latitude and wind velocity and show them to be in good agreement with other studies. Differences between studies can largely be explained by different data sets, time periods, pressure level and/or methodology (Pena-Ortiz et al., 2013; Rikus, 2015)."

p. 13, l. 1-5: added paragraph "Parameters for the third dimension could be optimized in a similar way as done for latitude, but using pressure heights.

In addition, to account for splitting of the STJ and PFJ, we plan to calculate not two, but four (or even more) jet stream cores with different climatological mean latitude  $\phi_{clim}$ . In cases, where only one path exists, the found jet stream cores would be combined to one path (based on their similarities to each other) and in other cases, where two paths exist, they would split. "

p. 14, l. 12-14: added reference: Di Capua, G. and Coumou, D.: Changes in meandering of the Northern Hemisphere circulation, Environ. Res. Lett., 11(9), 94028, doi:10.1088/1748-9326/11/9/094028, 2016.

p.14, l. 16 – 19: changed reference to: "Coumou, D., Petoukhov, V., Rahmstorf, S., Petri, S. and Schellnhuber, H. J.: Quasi-resonant circulation regimes and hemispheric synchronization of extreme weather in boreal summer, Proc. Natl. Acad. Sci., 111(34) doi:10.1073/pnas.1412797111,2014."

p. 14 |. 30- p. 15, l. 1: changed reference to : "Flechsig, M., Böhm, U., Nocke, T. and Rachimow, C.: The Multi-Run Simulation Environment SimEnv, , 1, [online] Available from: https://www.pikpotsdam.de/research/transdisciplinary-concepts-and-methods/tools/simenv/, 2013."

p. 15, l. 2 - 3: changed reference to: "Grise, K. M. and Polvani, L. M.: The response of midlatitude jets to increased CO2: Distinguishing the roles of sea surface temperature and direct radiative forcing, Geophys. Res. Lett., 41, doi:10.1002/2013GL058489,2014."

p. 15, l. 12-13:added reference: Kornhuber, K., Petoukhov, V., Petri, S., Rahmstorf, S. and Coumou, D.: Evidence for wave resonance as a key mechanism for generating high-amplitude quasi-stationary waves in boreal summer, Clim. Dyn., doi:10.1007/s00382-016-3399-6, 2016.

p. 15, l. 20-21: added reference: "Limbach, S., Schömer, E. and Wernli, H.: Detection, tracking and event localization of jet stream features in 4-D atmospheric data, Geosci. Model Dev., 5(2), 457–470, doi:10.5194/gmd-5-457-2012, 2012."

p.16, l. 1 - 2: changed reference: Pollard, E. and Lakhani, K. H.: The Detection of Density-Dependence from a Series of Annual Censuses Author, Ecology, 68(6), 2046–2055, 1987.

p.16, I. 5-6: changed reference: Schreiber, T. and Schmitz, A.: Surrogate time series, Phys. D Nonlinear Phenom., 142(3–4), 346–382, doi:10.1016/S0167-2789(00)00043-9, 2000.

p. 18, l. 1-4: changed Fig. 3 to Fig. 2

p. 18, I. 7-11: changed Fig. 4 to Fig. 3 and changed "white points" to "blue points"

p. 19, l. 1-5: changed Fig. 2 to Fig. 4

p. 20, l. 1-5: removed "compare Fig. 2"

p. 22, l. 2-5: removed annual probability density plot.

p. 26, l. 1: added an additional figure for global mean trends: Fig. 14: Annual, DJF, and JJA: Mean latitudinal trends and mean wind velocity trends of the STJ and PFJ cores.

[revised manuscript text omitted]

---

## Editor Decision (ED1)

**ESD-2016-37: Editor Decision Letter**

November 7th, 2016

Dear authors,

Thank your very much for your comments and developments in response to the reviewer reports.

The scientific questions addressed in the manuscript are relevant to the scope of Earth System Dynamics, and significant efforts have been made by the authors in addressing such questions.

However, the manuscript would considerably benefit from further elaboration, clarification and revision in order to address the concerns raised in the peer-review process.

In this regard, the diligences conducted in response to the reviewers are already an encouraging step forward.

At this stage, the authors are then encouraged to proceed with their review efforts, with particular emphasis on the reviewer recommendations.

In order to provide enough time for a careful revision further harnessing the valuable potential of this manuscript, my decision entails "major revisions".

I will be looking forward to the revised manuscript.

With very best wishes,

Rui Perdigão
(ESD Editor)